# Near Real-Time Automated Classification of Seismic Signals of Slope Failures with Continuous Random Forests

Michaela Wenner[1,2], Clément Hibert[3], Alec van Herwijnen[4], Lorenz Meier[5], and Fabian Walter[1]

[1]Laboratory of Hydraulics, Hydrology and Glaciology (VAW), ETH Zurich, Zurich, Switzerland
[2]Swiss Federal Institute for Forest, Snow and Landscape Research, Birmensdorf, Switzerland
[3]Université de Strasbourg, CNRS, EOST/IPGS UMR 7516, F-67000 Strasbourg, France
[4]WSL Institute for Snow Avalanche Research SLF, Davos, Switzerland
[5]Geopraevent Ltd., Zurich, Switzerland
**Correspondence:** Michaela Wenner (wenner@vaw.baug.ethz.ch)

**Abstract.** In mountainous areas, rockfalls, rock avalanches, and debris flows constitute a risk to human life and property. Seismology has proven a useful tool to monitor such mass movements while increasing data volumes and availability of real-time data streams demand new solutions for automatic signal classification. Ideally, seismic monitoring arrays have large apertures and record a significant number of mass movements to train detection algorithms. However, this is rarely the case, as a result of cost and time constraints and the rare occurrence of catastrophic mass movements. Here, we use the supervised random forest algorithm to classify windowed seismic data on a continuous data stream. We investigate algorithm performance for signal classification into noise (NO), slope failure (SF), and earthquake (EQ) classes and explore the influence of non-ideal though commonly encountered conditions: poor network coverage, imbalanced data sets, and low signal-to-noise ratios (SNR). To this end we use data from two separate locations in the Swiss Alps: data set (i), recorded at Illgraben, contains signals of several dozen slope failures with low SNR, data set (ii), recorded at Pizzo Cengalo, contains only five slope failure events albeit with higher SNR. The low SNR of slope failure events in data set (i) leads to a classification accuracy of 70% for SF, with the largest confusion between NO and SF. Although data set (ii) is highly imbalanced, lowering the prediction threshold for slope failures leads to a prediction accuracy of 80% for SF, with the largest confusion between SF and EQ. Standard techniques to mitigate training data imbalance do not increase prediction accuracy. The classifier of data set (ii) is then used to train a model for the classification of 176 days of continuous seismic recordings containing four slope failure events. The model classifies eight events as slope failures, of which two are snow avalanches, and one is a rock slope failure. The other events are local or regional earthquakes. By including earthquake detection of a permanent seismic station at 131 km distance to the test site into the decision-making process, all earthquakes falsely classified as slope failures can be excluded. Our study shows that even for limited training data and non-optimal network geometry, machine learning algorithms applied to high-quality seismic records can be used to monitor mass movements automatically.

# 1   Introduction

High mountain areas are particularly affected by climate change. Deglaciation and thawing of permafrost has implications on rock wall stability at high elevation and, consequently, on communities down-valley (e.g., Allen and Huggel, 2013; Phillips et al., 2017; Coe et al., 2018; Hock et al., 2019). The increasing threat to mountain communities, especially in densely populated areas, demands new monitoring techniques at high temporal resolution and broad spatial coverage to improve predictability, alarm time and post-event intervention. As a result of incomplete data and knowledge on relevant processes and triggering mechanisms, accurate prediction of rockfall events is still not possible (van Westen et al., 2006). However, an increase in slope activity (pre-event acceleration and increased frequency of small events) is a possible precursor to larger destructive events (Rosser et al., 2007). Existing methods to monitor slope failures include point measurements (e.g., extensometers) and large scale monitoring such as terrestrial laser scanners, interferometric radar, and video image recognition (e.g., Abellán et al., 2011). However, these techniques suffer from disadvantages like high operating costs, limited spatial coverage, and susceptibility to atmospheric conditions.

In the last decade, seismology has evolved into a method to monitor earth surface processes. Knowledge from wave propagation within the earth and generation mechanisms of seismic waves is transferred from its original study objectives, earthquakes, to the so-called field of environmental seismology (e.g., Burtin et al., 2008; Deparis et al., 2008; Helmstetter and Garambois, 2010; Gimbert et al., 2014; Hibert et al., 2014; Larose et al., 2015; Dietze et al., 2017; Allstadt et al., 2018; Lai et al., 2018). Seismic signals generated by mass movements are typically emergent with dominant frequencies of $5 - 10$ Hz and few or no distinguishable seismic phases. Signals of large events ($> 10^5$ m$^3$) recorded at far distances are characterized by long-period seismic waves ($< 0.1$ Hz) (e..g., Allstadt, 2013). Signal duration varies between seconds and several minutes, depending on the type of slope failures and slope scales (e.g., Vilajosana et al., 2008; Hibert et al., 2011; Dietze et al., 2017).

Seismometers can record large mass movements up to hundreds of kilometers away from the source (e.g., Allstadt, 2013; Walter et al., 2020) and allow continuous monitoring of large areas with real-time data transmission (e.g., Ekström and Stark, 2013). Additionally, the installation of seismometers is relatively low-cost and straightforward as no high-power supply and little or no or construction is needed, which is required, e.g., for interferometric radar. On the other hand, seismic sensors are sensitive to various sources like earthquakes, anthropogenic noise, atmospheric signals, runoff, and slope instabilities. Consequently, to distinguish signals of slope failures from other mechanisms, automated techniques for detection and classification are needed.

One approach that is often used to detect seismic signals is the well established short-term average over long-term average (STA/LTA) detection method, based on signal amplitudes (Allen, 1982). However, to classify source mechanisms, information on signal frequency content is often also required. For signals with similar frequency content, amplitudes and signal duration, such as earthquakes and slope failures, detection of signals from only one source mechanism with STA/LTA is therefore impossible. Additionally, parameter selection for optimizing STA/LTA is a tedious process that requires detailed knowledge of the data. Furthermore, seismic signals of slope instabilities are characterized by an emergent onset, making detection with STA/LTA difficult. For this reason, Helmstetter and Garambois (2010) suggested an STA/LTA algorithm for use in the fre-

quency domain that reliably detects seismic signals without impulsive onsets (e.g., Hibert et al., 2017). Nevertheless, the detector does not allow a distinction between different generation mechanisms. Therefore, Hibert et al. (2017) and Provost et al. (2017) used a supervised machine learning algorithm, random forest, to automatically classify local events detected with the adapted STA/LTA algorithm of Helmstetter and Garambois (2010). High accuracy (99 % and 93 %, respectively) emphasizes such algorithms' capability to classify seismic signals.

One downside of the combined STA/LTA approach by Hibert et al. (2017) and Helmstetter and Garambois (2010) is that this method requires two optimization steps - choosing the correct parameters for the STA/LTA algorithm and for the classifier. For this reason, this approach is called the two-step approach in the further course of the manuscript. Moreover, STA/LTA algorithms generally fail to detect signals which emerge over a timescale larger than the long term average window.

As an alternative to the two-step approach, a stochastic classifier, hidden Markov models (HMMs), has been used to automatically detect and classify a variety of seismic sources (e.g., Hammer et al., 2013; Dammeier et al., 2016; Heck et al., 2018). Hammer et al. (2013) and Dammeier et al. (2016) focused on a regional scale with larger rockfall volumes ($> 1000 \ m^3$) detected tens to hundreds of kilometers away from the source. They show that HMMs successfully classify seismic signals on a continuous data stream. However, to minimize false detection and misclassification, careful retraining and post-processing steps were required. Dammeier et al. (2016) compared the classification output with an earthquake catalog and suggested that when using HMMs in an actual operational setting, the on-duty operator should manually inspect the signal and decide if the event is an earthquake or a slope failure. Yuan et al. (2019) used random forest to classify seismic signals in several minute-long windows of seismic data. The study focuses on multiple days of data recorded near a geyser to detect pre-eruption seismicity, which is hidden in the noise. In the following, signal classification on the continuous data stream that does not require separate event detection is called the one-step approach.

In this paper, we use the random forest algorithm (Breiman, 2001) to perform automatic signal classification on continuous data on a local scale and throughout an extended time period with a large variety of noise signals. Previous local-scale two-step approaches have used specialized networks designed to maximize monitoring capabilities (e.g. Provost et al., 2017). However, due to logistical and financial constraints, this is often not possible for potential hazard sites. Here, we compare the one-step approach's performance in non-ideal conditions applying it to a data set with (i) many small slope failure events with low signal-to-noise ratio (Illgraben) and (ii) few events but a higher signal-to-noise ratio (Pizzo Cengalo). Furthermore, we show that by adjusting our methodology to a network with a sub-optimal configuration and a data set with only a few recorded events available for training, automatic detection of potential slope failures is still possible.

The scope of this study is to a) test a system of continuous classification of windowed seismic data on two different types of data sets, b) test the influence of signal to noise ratios and an imbalanced training data set on classifier performance, c) discuss insights on the transferability of trained classifiers to other sites, and d) mimic operational conditions to assess our approach's capability as an alarm system for slope failures. The investigation of source mechanisms and processes of seismogenic mass movements is outside our study's scope. Such an endeavor should not be based on weak seismic signals, which make up a large part of our catalog.

## 2 Study Sites and Data Set

### 2.1 Illgraben, Switzerland

The Illgraben catchment in southwest Switzerland is one of the most active mass wasting sites in the European Alps (Fig. 1a). Yearly precipitation is controlled by summer rainstorms with high rainfall intensity during which mass wasting with rock-slope failures and debris flows occur regularly (e.g., Badoux et al., 2009). From its highest point at the Illhorn (2716 m.a.s.l), the Illgraben catchment reaches down to the Rhone Valley (600 m.a.s.l), where its main torrent flows into the Rhone River. The 9.5 km$^2$ Illgraben catchment is characterized by complex geology, where highly fractured quartzite is the dominating bedrock at the north-west face of the Illhorn and the head of the channel trunk and limestone at the southeast facing slope of the catchment (Schlunegger et al., 2009; Bennett et al., 2013). The fractured quartzite with erosion rates of tens of centimeters per year is the main contributor to sediments transported via debris flows (Bennett et al., 2013). A seismic network of 8 high-quality stations was installed throughout Illgraben between May and September 2017 to monitor rock-slope as well as debris-flow activity. Walter et al. (2017) used seismic data from a similar array to locate a debris-flows front as it propagated along the channel. In this study, we focus on slope activity at the Illhorn north-west face, which at peak times is characterized by several slope failures per day. Slope-failure volumes are to date not quantified, but direct field observations by the authors indicate volumes of tens to hundreds of cubic meters. We use seismic data recorded by three seismic stations (ILL06, ILL07, ILL08) in 2017, located closest to the area of interest (Fig. 1a). The three-component seismometers (LE-3Dlite) with a lower cutoff frequency of 1 Hz and a sampling frequency of 100 Hz were installed with a mean inter-station distance of about 1 km and hundreds of meters from the Illhorn north-west face.

### 2.2 Pizzo Cengalo, Switzerland

Pizzo Cengalo is a mountain located in Val Bondasca in Eastern Swizerland's canton of Grisons (Fig. 1b) about 6 km south east of the down-slope village of Bondo near the Italian border. Pizzo Cengalo's slopes have been unstable for several decades, with multiple rock slope failures per year. Bergell Granite defines Pizzo Cengalo's geology and part of its north facing walls are covered by glaciers (Baer et al., 2017). After a large failure ($\sim 1.5 \times 10^6$ m$^3$) in 2011, systematic monitoring started in 2012 (Baer et al., 2017). In 2017, an even larger rock avalanche ($> 3.5 \times 10^6$ m$^3$) killed eight hikers and resulted in a series of debris flows that destroyed parts of the village of Bondo (Walter et al., 2020). A warning had been issued weeks prior to the catastrophic failure because an acceleration of slope displacement was observed, as well as several smaller failure events before the rock avalanche. The large event in 2017 prompted an extension of the monitoring system, which included the installation of three seismometers (LERA1 – 3) close to the Bondasca river, the outlet of the catchment, some 3.5 km down-valley of Pizzo Cengalo and 2.5 km up-valley of Bondo.

The one component, short-period geophones (GeoSig 0.9 Hz), with a flat response from 0.9 Hz to 89 Hz and a sampling frequency of 200 Hz, were installed along the channel with a mean inter-station distance of about 20 m (Fig. 1b). An array in such a configuration can be used to detect debris flows based on amplitude differences while the flow approaches, passes by, and moves away from the stations (Coviello et al., 2019). Since the installation of the seismometers, seismic signals and

spectrograms have been made available to stakeholders in hourly time windows in the online portal of the engineering company Geopraevent, to recognize an increase in rockfall activity by visually evaluating the seismic data. Until now, this required daily visual inspection of the seismic signals by employees of the canton of Grisons, who are not trained seismologists.

In November 2017, we installed an automatic camera facing Pizzo Cengalo (768411 / 132790, Fig. 1b). Images were taken every 30 minutes and transferred in real-time via a cellular connection. We used these images to validate detected events, and the images were also available to stakeholders on the online portal of Geopraevent.

## 2.3 Labeled Data Set

For supervised machine learning algorithms, a set of labeled data has to be provided. Here, we focus on seismic data from 3 sensors of the ILL array recorded in 2017 and the 3-sensor LERA array recorded in 2018 and 2019. We compiled an event database for both study sites by visual inspection of the seismic waveforms and spectrograms of seismic stations at Illgraben and Pizzo Cengalo and close-by stations of the Swiss Seismological Service (SED) (Illgraben: Stations CH.LKBD, CH.VANNI; Pizzo Cengalo: Stations XP.PICE1, CH.VDL, CH.FIESA). Additionally, we use a list of observed slope failures made available by the canton of Grisons for Pizzo Cengalo and earthquake catalogs from SED and the European-Mediterranean Seismological Centre (EMSC). For the classification, we decided to use three different classes: noise (NO), slope failures (SF), and earthquakes (EQ). The NO class contains samples of continuous noise and noise signals of anthropogenic and atmospheric origin (Marchetti et al., 2019). We use the SF class as an umbrella term for all types of mass movements that might occur (e.g., snow/debris avalanches, rockfalls). We consider this assumption valid, as different granular flow types share the common seismogenesis of particle ground impacts (e.g., Suriñach et al., 2005; Farin et al., 2019), although length and amplitudes depend on runout distance and volume. We assume that differences in signal characteristics between EQ, NO, and SF class are more significant than discrepancies between different types of granular flow signals. The most critical parameters to manually classify an event as slope failure are: dominant frequencies of 5-10 Hz, emergent onset, no phase arrivals, not listed in earthquake catalog, not seen on surrounding seismic stations, or lower amplitude at surrounding stations.

The EQ class contains a set of local, regional, and teleseismic events. Extensive testing showed that lumping all earthquakes in one class does not negatively affect classifier performance (Fig. A2). An example signal of each class is presented in Fig. 2. For the continuous noise, we choose random times over the year. For noise signals, earthquake signals, and slope failure signals, we manually picked the events' start-time and end-time when the signal exceeds the noise level. The number of events in each class for both study sites is presented in Fig. 3a and Tab. B1. Note that there are only five events that are related to slope failures at Pizzo Cengalo in 2018. Due to instrument malfunction, these events were not captured by the automatic camera. The sparsity of recorded events leaves us with a poor data set for this class. This issue is addressed further in section 3.2. Figure 3b shows a boxplot of the signal to noise ratios of SF and EQ events at both sites. The signal-to-noise ratio for earthquakes ranges between 1 and $10^3$, with a mean of 5 (Fig. 3b). The signal-to-noise ratio for slope failures varies between 1 and 30, with a mean of 4 for the Illgraben data set and 22 for the Pizzo Cengalo data set.

## 3 Methodology

A schematic illustration of our automatic one-step classifier applied to the Pizzo Cengalo data set is shown in Fig. 4a. We use random forest, a supervised ensemble machine learning algorithm (Breiman, 2001), to classify different seismic sources using a running window on a continuous data stream on all network stations. Random forest is based on the majority vote of several weak decision trees, where each decision tree is built on a random subset of features and training data set. Decision trees consist of nodes, branches, and final nodes. A split based on a threshold on a variable is performed on each node, resulting in one or two branches. This process continues until a classification result is obtained in a final node, a so-called leaf. A single weak tree performs poorly in the classification task, as it is only trained on a subset of features and the training data set. However, the performance improves as the aggregated decision trees perform a majority vote, where the proportion of trees that voted for one class gives the probability for the class. The time window is then labeled according to the class with the most votes, i.e., the highest probability. We choose random forest, because (i) it is a comprehensive machine learning algorithm that has shown to outperform other algorithms, like support vector machines and boosting ensembles, in a variety of cases (Fernández-Delgado et al., 2014) and (ii) it already was successfully used to classify rock slope failures (Hibert et al., 2017; Maggi et al., 2017; Provost et al., 2017; Malfante et al., 2018; Hibert et al., 2019). Moreover, random forest estimates the feature importance by measuring the impurity, which describes how many samples of how many different classes belong to one node after a split. Hence, if all samples in a node belong to one class, impurity is zero, and the classifier is perfect. The averaged impurity decrease from a feature over all decision trees then gives a ranking of the most discriminating feature. This allows a more detailed analysis of potential causes for misclassification. For the implementation of random forest we use scikit-learn, a python library for machine learning (Pedregosa et al., 2011).

### 3.1 Data Stream Handling

To avoid the extra step of detecting events with a trigger such as the STA/LTA algorithm, we classify a running window on the continuous data stream with an overlap of $2/3$ of the window length. The overlap was chosen to avoid missing events on the window margins but was not tested for optimal performance. We transform the event catalog with start-times and end-times of all events into a catalog containing the times of all running windows that include an event for both the Illgraben and the Pizzo Cengalo data set. For the Pizzo Cengalo data set, we make use of the network configuration at the study site, in order to increase the number of training samples. At the frequency band of interest (1-10 Hz), associated wavelengths are larger than the inter-station distance, resulting in waveforms with only small differences. For earthquakes and slope failures, instead of using the same onset for the sliding windows on all stations, we choose a random onset with a maximum of $2/3$ of the sliding window before the event start-time. This way, we catch different windows of the signal and increase the training data set by a factor of three without using the same window several times. This procedure is unnecessary for the Illgraben data set, as the source-receiver distances vary for each station. For discrete noise signals often recorded on only one station, we choose sliding windows on the recording station, again with a random onset up to $2/3$ of the sliding window before the event start-time. For the continuous noise, we choose a random station at each time step.

We divide the catalog with labeled events into a training and test data set, with 70% of all events as training data and 30% as test data. This partition was chosen to be able to meaningfully assess the algorithm performance for the small number of slope failures in the Pizzo Cengalo data set. This way, there are windows of three slope failure events in the training and validation data set and windows of two slope failure events in the test data set. We choose a window length of 20 seconds as an initial guess. Numbers for resulting training and test data set sizes for Pizzo Cengalo and Illgraben are presented in Tab. B1. Following Provost et al. (2017), we then compute features of these sliding windows. As we do not use the entire waveform of the event, but only the parts that appear in the sliding window, we exclude features related to the entire waveform of the signal (e.g., duration and rise time). Additionally, the network configuration at Pizzo Cengalo does not allow network features (signals are too similar between stations), nor does it allow for polarity features (only vertical component available). For the sake of comparison, we also disregard network and polarity features for the Illgraben data set. We are left with a total number of 55 features, including waveform characteristics in the time and frequency domain (see Table A1). These features have been proven significant for accurate seismic signal classification (e.g., Hibert et al., 2017; Provost et al., 2017). Additionally, we tested a python tool for automatic feature generation for time series (TSFRESH, Christ et al. (2016)), which did not improve classification results compared to the features proposed by Provost et al. (2017). Before feature calculation, we apply a four corners Butterworth bandpass filter $(1-10\,\mathrm{Hz})$. For feature generation after Provost et al. (2017), we choose frequency bands of $1-3\,\mathrm{Hz}$, $3-6\,\mathrm{Hz}$, $5-7\,\mathrm{Hz}$, $6-9\,\mathrm{Hz}$, and $8-10\,\mathrm{Hz}$.

### 3.2 Imbalanced Data Set Handling

The limited amount of data and, more specifically, the small number of SF events that happened in 2018 at Pizzo Cengalo lead to an imbalanced data set. As shown in Fig. 3a, the number of events is unevenly distributed among classes. This poses a problem for machine learning algorithms, as they generally optimize the score, i.e., the number of correctly labeled classes. In a highly imbalanced data set, the classification algorithm may be less sensitive to the minority class, as it does not drastically impair the score if it is labeled incorrectly. For our data set, with the events that we are most interested in being the minority class, it is particularly important to address this problem. Therefore, we introduce here different data augmentation and classifier tuning methods to improve the classifiers' performance on the imbalanced Pizzo Cengalo data set.

There are several possibilities to handle imbalanced data sets, either based on manipulating the training data set or on changes within the algorithm (Chawla, 2010). The most straightforward approaches are random undersampling (US) and naive oversampling (OS) training data. For random undersampling, only a random subset of training data of the majority class is chosen. This way, the data set becomes more balanced by reducing the samples in the majority classes. However, this might mean that important characteristics of the majority class are not captured. In contrast to undersampling, naive oversampling randomly multiplies samples in the minority class but thus increases the risk of overfitting, the lack of generalization, within the minority class.

A more sophisticated way of increasing training samples in the minority class is synthetic minority over-sampling (SMOTE) (Chawla et al., 2002). SMOTE is based on the idea of creating new training samples in the minority class by interpolating between a sample and a random set of its k-nearest neighbors in feature space. Therefore, a new sample is generated with

features similar to already existing samples. This increases the sample size of the minority class but minimizes the problem of overfitting. On the algorithm level, random forest opens two possibilities for imbalanced data: setting a class weight on the minority class or undersample the training data for every single tree, a so-called balanced random forest (BRF) (Lemaître et al., 2017). Here, we use a BRF classifier that undersamples the majority class and weighs the classes inversely to the number of samples in each class.

### 3.3 Training Process and Evaluation

We use two different metrics to evaluate model accuracy, the confusion matrix and receiver operating characteristic (ROC) curves. The confusion matrix consists of the true label of each class's samples as rows and the classifier predicted label as columns (Fig. 4b). For a perfect classifier, all samples are located on the diagonal of the matrix. Using the confusion matrix, the classifier can be evaluated for each class separately. Furthermore, we normalize the confusion matrix, such that the sum of each row is 1.0. The ROC curve uses the true positive rate (TPR) and false positive rate (FPR) for different probability thresholds (Fawcett, 2006). TPR is defined as the number of true positives divided by the sum of true positives (TP) and false negatives (FN) (TPR = TP/(TP + FN)). FPR is defined as the number of false positives (FP) divided by the sum of false positives and true negatives (TN) (FPR = FP/(FP + TN)). Class prediction of random forest is based on the score of a class, i.e., its probability defined by the number of predictions out of all trees. By lowering the threshold for classification, i.e., the probability threshold for a class to be predicted, FPR and TPR increase as FN samples transition to TP and TN samples transition to FP. As an example, we consider an imaginary two-class problem with a decision threshold of probability $> 0.5$ for the "positive" class leading to a TP = 1, FP = 2 , TN = 3 and FN = 4 and resulting TPR = $1/5$ and FPR = $2/5$. When lowering the probability threshold for the "positive" class to be predicted to, say, 0.2, TP will increase, but so will FP, giving TP = 4, FP = 3, TN, = 2, and FN = 1. This results in larger values of TPR and FPR (TPR = $4/5$, FPR = $3/5$). When plotting FPR against TPR for each probability threshold, one obtains the ROC curve with a monotonous increase. For a schematic drawing, see Fig. 4b). The best-case scenario is a TPR of one, and an FPR of zero, i.e., a step transition from coordinates (0,0) to (0,1), whereas a random classifier would result in a diagonal from (0,0) to (1,1). The area under the curve (AUC) can be used as a one value metric for model performance. The larger the area under the ROC curve, the better the model accuracy. Both metrics, confusion matrix and ROC/AUC, can directly be transferred into a multiclass environment. For the confusion matrix, this simply results in several columns and rows. The ROC curve can be computed for each class separately by bundling all other classes together.

We use the confusion matrix as an initial performance evaluation of random forest for both the Illgraben and the Pizzo Cengalo data set. We then use the ROC/AUC analysis to estimate the influence of the window length and the imbalanced data set handling techniques on the Pizzo Cengalo data set. For a more representative measure, we use k-fold cross validation when computing the ROC curves (e.g., Stone, 1974). As our training data set only contains three events in the minority class, we use 3-fold cross validation, with random 2/3 of each class in the training data set and 1/3 used for validation. This way, we obtain three ROC curves and AUC values per class trained and tested on three different random subsets. We then take the mean TPR and FPR to plot the ROC curves. We computed the 95% confidence level using Student's $t$-distribution for small sample sizes (n=3).

## 3.4 Simulated Real-Time Monitoring at Pizzo Cengalo

To test how the classifier performs in a real-life application, we use a model trained on the 2018 Pizzo Cengalo data set and classify more than a million time windows of recorded seismic data at Pizzo Cengalo in 2019. We train, validate, and test the model on 2018 data, containing five slope failure events, and then use the model to classify 2019 data. Seismograms and spectrograms of the training events are shown in Appendix A1. For 2019, no full event catalog is available. We cross-check as slope failure classified events with earthquake catalogs, hiker reports, and pictures from the automated camera. If none of these methods give clarity, we manually classify the seismic signals based on typical characteristics of slope-failure events, such as dominant frequencies between 5 - 10 Hz, an emergent onset, and a duration of several tens of seconds (e.g., Hibert et al., 2011).

We benchmark our test with the two-step approach of STA/LTA detection in the frequency domain and classification of the detected events using random forest. After extensive testing, we define the parameters that provide accurate detection for our data set as an STA window length of 1s and an LTA window length of 18s. The detector turns on when the STA/LTA ratio exceeds four and turns off when the STA/LTA ratio becomes lower than two. Additionally, we use a coincidence trigger, with a threshold of three, which means that the STA/LTA threshold needs to be exceeded at all three stations.

## 4 Results

### 4.1 Classifier Performance on Labeled Data Sets

For an initial evaluation, we tested the random forest classifier's performance on the Illgraben and the Pizzo Cengalo data set. We used a randomized grid search to obtain the best performing hyper-parameters such as number and depth of decision trees. The final parameters are presented in Tab. B2. Normalized confusion matrices for both data sets are shown in Fig. 5. We trained the models on 20s window sizes, and no class balancing has been applied. For the Illgraben data set, the classifier correctly classifies between 70% and 85% for all classes. The largest confusion occurs between the NO and SF class. For the Pizzo Cengalo data set, the classifier correctly classifies between 90% and 100% for the NO and EQ class, but only 44% of the SF class. In this case, the largest confusion occurs between the SF and EQ class.

### 4.2 ROC Analysis on Pizzo Cengalo Data Set

We computed ROC curves and AUC values for different window sizes and under and oversampling techniques for the Pizzo Cengalo data set (Fig. 6). Figure 6a shows the AUC values for the SF class plotted in a heatmap. Rows show different window sizes (10s - 60s) and columns different techniques. Associated standard deviations are shown in Fig. 6b. Darker colors mark a larger AUC value and a smaller standard deviation. The different color maps highlight the difference between the AUC values and AUC standard deviations. Although almost all values lie within the confidence intervals, overall, the smallest window size and the largest window size give slightly better values with a small standard deviation. Additionally, the AUC value for a simple random forest and balanced random forest (BRF) is higher without the data manipulation techniques.

As a compromise between large AUC value and small standard deviation, we choose 40-second windows and classical random forest, i.e., without modifications for handling imbalanced data. The ROC curve of this configuration is shown in Fig. 6c. To make sure we correctly classify most of the windows containing a slope failure signal, we set the target TPR to $> 0.9$. From the 3-fold cross-validation ROC analysis, we obtain a mean probability threshold of 0.23 for a TPR $> 0.9$ in the SF class. As a next step, the optimal model parameters (i.e., number of decision trees, number of features chosen for each tree, maximum tree depth, ...) for the 40 seconds window size and random forest are chosen using the randomized cross-validation search of scikit-learn (Pedregosa et al., 2011). We use the obtained classifier to classify the test data set from the 2018 data, containing two SF events, which were not used for the model set-up and, therefore, an unbiased evaluation of the classifier. We set the probability threshold for the SF class to 0.23 as obtained from the ROC curve. Consequently, for probabilities higher than 0.23 for the SF class, the window will be classified as slope failures, even if another class has a higher probability.

### 4.3 Classifier Performance with Classification Threshold on Pizzo Cengalo Data Set

We used the low probability threshold from the ROC analysis to label all 40 seconds time windows of the Pizzo Cengalo test data set. The results of this set-up are shown in Fig. 7. The normalized confusion matrix (Fig. 7a) shows a misclassification of 20% for slope failures. Additionally, 10% of earthquakes are classified as noise. The misclassification rate of noise is however very small (1%). The most discriminating features are presented in Fig. 7b. The colors denote spectral and waveform features. Distinctive features are spectral gyration radius (gamma2), spectral centroid (gamma1), central frequency of the first quartile (Fquart1), variance of the normalized fast Fourier transform (FFT) (VarFFT), frequency at the maximum of the FFT (FmaxFFT), frequency at spectrum centroid (FCentroid), energy of the last 2/3 of the autocorrelation function (INT2), and the energy of the seismic signal in the frequency band of 1-3 Hz (ES[0]) (Tab. A1, Provost et al., 2017). Figure 7b) shows that the by far most discriminating features are characteristics in the frequency domain.

### 4.4 Classifier Implementation on Pizzo Cengalo Seismic Data

As a next step, the model is used to classify seismic data recorded at Pizzo Cengalo in 2019, mimicking operational conditions of a near real-time classification. We first compute the signal features of 40 second time windows with an overlap of 2/3 of the window length for each station and perform a classification. Next, a majority vote of the stations is performed, and a label is assigned to the time window. This means that if more than one station assigns the time window to the same class, the end label is chosen accordingly. In case every seismic station classifies the same time window into a different class, the time window will be labeled as noise. We compare the results to an event catalog compiled from hiker reports, manual classification of seismic data and automatic camera images. The classification parameters for the manual classification of the seismic data are consistent with creating the training catalog.

In 176 days in 2019 (Julian Day 94 to 270), 21 days have at least one window classified as slope failure. To exclude misclassified windows because they only contain a small signal portion, we set a minimum threshold of three consecutive SF classifications. With this threshold, we limit the number of slope failure detections to eight. Out of these eight, three correspond to manually picked slope failures, two of which happened on 4 and 26 April and one on 16 July. However, the automatic camera

shows that the April events are snow avalanches rather than rock slope failures (Fig. 8). Seismic waveforms and associated classifications for 16 July are shown in Fig. 9, with a zoom on a rockfall event (c-e) and a noise signal (f-h). Four events that were classified as slope failures are earthquakes on 22 May, 29 July, 8 August and 29 August. Two out of these earthquakes originate from the German lakeside of Lake Constance about 160 km north-west of Pizzo Cengalo, with a distance between the epicenters of about 3 km and magnitudes of 3.6 and 3.4 (EMSC catalog). The two other earthquakes are Magnitude 3.3 and 2.2 earthquakes with epicenters about 170 km south and 33 km west of Pizzo Cengalo, respectively (EMSC catalog). The last event classified as slope failure on August 13 is characterized by a duration of 10 s and is not listed in any earthquake catalog. All waveforms, spectrograms, and spectra are shown in Fig. 10.

To quantify the performance of the algorithm presented here, we benchmark the continuous approach against a two-step approach with an STA/LTA detection on the Pizzo Cengalo data set. The continuous approach correctly classifies three slope failures (TP), misses one slope failure (FN), and classifies four earthquakes as slope failures (FP). The two-step approach of STA/LTA detection correctly classifies two slope failures (TP), misses two slope failures (FN), and classifies six earthquakes as slope failures (FP). For a simple comparison, we can use the critical success index (CSI = TP / (TP + FN + FP)) which ignores all non events (TN). For the Pizzo Cengalo data set, we obtain a CSI of 0.375 for the continuous approach, whereas, for the STA/LTA approach, we obtain a CSI of 0.2.

### 4.5   Transferability to other study sites

We tested the transferability of a trained model to other study sites by assessing the performance of a model trained on one site and tested on another, and trained on both sites and tested on one site. The result of all possible combinations is shown in Figure 11. A classifier trained on one site and tested on another reduces the mean score over each class by about 30 % for the Pizzo Cengalo and the Illgraben data set. Especially the score of the minority class, the slope failure class, is reduced to 20 % and even 0 % respectively. For both the Illgraben and the Pizzo Cengalo data set, a classifier trained on both data sets increases the mean test scores over all classes only marginally. However, in both cases, the slope failure class's test scores increase by nearly 10 % compared to a classifier only trained on the respective data set.

## 5   Discussion

### 5.1   Data Set Comparison

We trained a random forest classifier on two different types of data sets: the Illgraben data set, with a balanced abundance of all types of classes, and the Pizzo Cengalo data set, with a highly imbalanced number of events. Both the Illgraben and the Pizzo Cengalo data sets were aggregated by manual inspection of the seismic data and, for the Pizzo Cengalo data set, direct observations at the site. Even though both data sets have been carefully examined, a misclassification of events cannot be excluded. For earthquakes, a misclassification is highly improbable, as all events were cross-checked with publicly available earthquake catalogs. However, cross-checking databases for rockfall events are rare, especially for events that have not affected

infrastructure. Therefore, an inherent bias by manual labeling and falsely labeled events is possible but can hardly be avoided for these data sets.

The Illgraben data set contains only four weeks of manually labeled data, but in those four weeks, the number of recorded slope failures and earthquakes is nearly the same (Fig. 3a). The large number of slope failure events during that period can be explained by three extreme precipitation events that also triggered three debris flows (e.g., Wenner et al., 2019). Under these conditions, slope failure activity increases drastically.

The Pizzo Cengalo data set contains manually labeled data of the summer period (June - October). The LERA network was set-up in the aftermath of a large rock avalanche event in 2017. Since then, slope activity has strongly decreased. This implies that automatic detection and classification is based on a small number of training events in the slope failure class and a comparably large number of events in other classes. The small number of recorded earthquakes can be attributed to general poor data quality. Many recorded earthquakes barely exceeded the noise level and were therefore not included in the catalog.

For slope failures, the Pizzo Cengalo data set includes fewer events than the Illgraben data set. However, the mean SNR is higher. Overall, SNR of events in both data sets compare to SNR found in other studies (e.g., Dammeier et al., 2016). The SNR distribution in both data sets is also reflected in the initial classification test (Fig. 5). In this test, no performance enhancement was applied. Whereas the classifier performs similarly well for each class for the Illgraben data set, 22% of the slope failure signals are misclassified as noise. The large number of misclassification could be related to the low SNR for slope failures in the data set. For the Pizzo Cengalo data set, the classifier performs significantly worse for the slope failure class, but the confusion is more pronounced between the slope failure class and earthquake class than slope failure and noise. Hence, the Pizzo Cengalo data set's misclassification might not result from poor SNR, but the underrepresentation of the slope failure class. We expect an improvement of the classifiers performance after adding slope failure seismic signals of future events to the training data set.

## 5.2 Performance Enhancement

One way to enhance the classifier's performance on the Illgraben data set could be to add network and polarity features. Provost et al. (2017) have shown that network features count among the most important features for classifying detected events. However, the set-up of the LERA network at Pizzo Cengalo has an unfavorable aperture to detect and classify detected seismic signals, as the stations are set-up in a line with interstation distances of $9\,\text{m}$ and $31.5\,\text{m}$. This prohibits the usage of network characteristics, like arrival time differences and amplitude ratios.

To address the problem of the imbalanced Pizzo Cengalo data set and resulting misleading scores, we test several techniques to handle such data sets and use receiver operating characteristic curves for performance assessment. The area under the curve for SF is largest for a generic random forest. Further assessment shows that SF's true positives are largest when using a technique to handle imbalanced data sets, but leading to a substantial increase in EQ events being classified as SF. Using a generic random forest, SF is underrepresented, ending up with zero true positives but also zero false positives. By lowering the probability threshold for SF, the true positive rate increases, whereas the false positive rate stays low. Therefore, for this data set, we decided to ignore problems with imbalanced data sets and mitigate misclassifications by lowering the probability

threshold. It remains to be seen if this approach performs equally well on other data sets, but in our case, it gives the best

results by maximizing the number of true positives in SF and minimizing the number of EQ classified as SF. This does not mean, however, that imbalance countermeasures did not work. Compared to the initial test (Fig. 5b), any of the measures mentioned above improve the classification result.

We choose to use a 40s window as an operative window size to test on the 2019 data, even though 10s and 60s show equally high AUC values and low standard deviations. On a continuous data stream a 10s window with 6s of overlap does

not leave enough time to compute features and classify the event in real-time. The 60s window, on the other hand, results in a classification delay of one minute and we assume that chances are higher to miss smaller events which are masked by a large amount of noise in the 60s window.

Generally, an imbalanced data set can be tackled by increasing the amount of training data in the minority class. A classifier trained for an area that is more active or has been monitored during a longer period is expected to give better results with

395 higher accuracy. Additionally, the small number of events in the slope failure class can lead to overfitting, i.e., an insufficient generalization of the model. Hence, small deviations in signal characteristics can lead to misclassification and undetected slope failures. However, as seen on tests on the Illgraben data set, solely a larger training data set will not give a perfect classifier either. We attribute the low SNR to the classifier's relatively poor performance of 30% misclassified events in the slope failure class at Illgraben.

**5.3   One-step vs. two-step method at Pizzo Cengalo**

Several studies have shown that classification algorithms accurately classify events detected with the STA/LTA approach (e.g., Hibert et al., 2017; Provost et al., 2017). The benchmark analysis performed on the Pizzo Cengalo data set indicates, that our continuous approach (CSI of 0.375) performs slightly better than a two-step approach (CSI of 0.2). However, the small number of events prohibits a statement on robustness. Interestingly, there is a large overlap in the earthquakes being misclassified as

slope failures between the two approaches.

Feature importance analysis for the continuous one-step approach presented here shows that the classifier predominantly uses spectral features to distinguish between different classes. This is consistent with the fact that the windowing eliminates information from the entire waveform. Provost et al. (2017) showed that for the two-step approach, several waveform features, e.g., duration and the ratio between ascending and descending time of the signal, are powerful distinctive features of slope

failures and earthquakes. These are, however, characteristics of the entire waveform of an event. In our case, constant window size with start and end regardless of event start and end sacrifices this information, and the classifier, therefore relies on spectral features. However, the spectral content of earthquakes and slope failures at our site is highly similar, which complicates a correct classification (Fig. 2). This is illustrated by the univariate distributions and correlations in Fig. 7c, that show large overlaps between the classes, even for the most discriminating features.

Despite the loss of several waveform features, the continuous approach outperforms the two-step approach in our case. We attribute the enhanced performance of the continuous approach to the reduced parameter tuning effort: In the two-step approach,

the performance of the classifier is strongly influenced by the STA/LTA detectors accuracy. Tests show that a manually picked catalog achieves up to a 15% higher accuracy than the same catalog compiled with an STA/LTA algorithm.

Even though not a focus of this study, we note that STA/LTA detection algorithms tuned to detect short signals (several tens of seconds) miss events of long duration and gradual amplitude increase, such as debris flows, volcanic tremors, lahars, and glacier lake outburst floods. Coviello et al. (2019) show that with a window size of 10s and 100s for STA and LTA, respectively, debris flows can be detected, excluding other events like earthquakes. However, this also excludes the detection of short slope-failure signals. The continuous approach is capable of detecting such events and is therefore applicable in multiple contexts and different sites. For example, intense precipitation raises the noise level by an increase in runoff and, consequently, seismogenic sediment transport (Tsai et al., 2012; Burtin et al., 2016). Similarly, snow cover and strong temperature fluctuations can affect the instrument itself and change the noise level. A preliminary implementation of a classifier with a fourth class called runoff trained on two days of increased water discharge (measured with gauges) found two additional days of peak discharge. Using the two step-method of STA/LTA, requires a second STA/LTA algorithm with its own parameters to detect these signals. Consequently, applying continuous random forest in different circumstances is potentially a low effort, as there is no need to fine-tune the detection algorithm while improving the overall results over the two-steps approach.

### 5.4   Test on 2019 Pizzo Cengalo data

Continuous random forest correctly classifies events in the test data with a high SNR. A slope failure that was observed by hikers on 14 August was classified as noise. Often events create dust clouds that are easily noticeable despite a small mass-movement volume. The misclassification likely results from a low SNR (Fig. 10), hence a probably relatively small volume of the event, as all windows containing the event were classified as noise. For most waveform and spectral features, especially the most discriminating ones (feature importance analysis), values of time windows containing the slope failure signal do not differ from that of windows that contain only noise. Filtering of the waveform to minimize noise is difficult, as the frequency band of the slope failure coincides with the primary noise. Furthermore, the automatic camera pictures show no apparent detachment zone (Fig. 8). This either means that the break-off happened outside of the camera field or validates our assumption of a small volume event.

Two of the events classified as slope failure are snow avalanches in April 2019 (Fig. 8). At this time, Pizzo Cengalo's active slope was partially covered by snow. As both avalanches happened during snowfall periods with obscured view from the automated camera to Pizzo Cengalo, the events were only validated by pictures several days after the actual event. The random forest classifier was only trained on data from the summer period, and therefore, snow avalanches were not part of the training data set. However, the seismic signature of snow avalanches is highly similar to those of other mass movements (e.g., Suriñach et al., 2001; van Herwijnen and Schweizer, 2011; Heck et al., 2018). This also validates your assumption that signal differences between different mass movements are less prominent than differences associated with noise and earthquakes. Hence, for training purposes, snow avalanches could be included in the slope failure class. This would allow the detection of slope failure events in winter and in summer month and increase the number of events in the slope failure class, resulting in a less imbalanced training data set.

We chose a minimal of three consecutive windows classified as slope failure for a slope failure event detection. If we were to increase the number to four, the number of slope failure event detections is lowered to four. Out of these four, only one is an actual slope failure and three are earthquakes. If we were to decrease the number of consecutive windows that need to be classified as slope failure to one, three more earthquakes would cause a slope failure event detection. Even by lowering the number of windows, the missed slope failure event would still not be detected. Thus, the classifiers' performance in correctly classifying three slope failure events is overshadowed by the four false alarms despite the consecutive window threshold. For operational use, a false alarm ratio $> 50\,\%$ is not acceptable as operators will lose confidence in the classifier.

In a promising approach, Hammer et al. (2017) use Hidden Markov Models to detect snow avalanches by training a background model from continuous seismic data and using seismic signals of only one avalanche to distinguish noise from avalanches. However, the model was only tested on five days of data. Continuing with this approach Heck et al. (2018) classify more than 100 days and find that for a reliable classification with a small number of false alarms, a daily update of the background model, as well as extensive array-based post-processing is necessary. With our approach, using random forest, no retraining of the background model is necessary and thus, the computational cost for a single parameter tuning is negligible. For operational use at the Pizzo Cengalo site, array-based post-processing as proposed by Heck et al. (2018) is not possible. Amongst other array-based post-processing steps Heck et al. (2019) include detections from a seismic station 14 kilometers away from the test site in the decision process to discriminate between noise signals and avalanches. We explore this approach with our classification problem of earthquakes falsely classified as slope failures. We train a model for a seismic station of the Swiss Seismological Service (CH.FIESA) located 130 km away from Pizzo Cengalo. At this distance local and regional earthquakes that were misclassified as slope failures at Pizzo Cengalo are still recorded, but not slope failures. We use the EMSC earthquake catalog to compile a training data set containing 45 earthquakes and 513 noise samples in 2018. We then train the model on 20s windows of seismic data. For the decision-making process, we test the trained model on seismic data recorded when a slope failure was classified at Pizzo Cengalo in 2019. All four earthquakes falsely classified as slope failures at Pizzo Cengalo were classified as earthquakes at FIESA. All time windows around the slope failures at Pizzo Cengalo were classified as noise at FIESA. Therefore, a simple inclusion of a station at a regional distance to Pizzo Cengalo reduced the false alarms to zero.

Our approach is computationally inexpensive, as the classification model only has to be trained once. From there on, computational power is only needed to compute features of the 40s windowed seismic data. For this near-real-time simulation, we used a standard machine (2017, Inter Core i7), which was able to compute all features and classify the windowed data within 3 seconds. Combined with the consecutive window threshold, this allows for a short warning time within tens of seconds after the event start

## 5.5 Classifier Transferability

For future applications, a key question is how a classifier trained on one data set can be transferred to other environments and study sites. Especially for sites previously not monitored, a classifier trained on other sites could give a head start in seismic monitoring. Figure 11 clearly shows that a classifier trained on one site and tested on another performs poorly. However, for

both the Illgraben and the Pizzo Cengalo data set, a classifier trained on both data sets increases the test scores for all classes, and especially the minority class, compared to a classifier only trained on the respective data set. Therefore, we suggest that cross-context training and data set amalgamation has the potential to improve classification outcome and should be investigated in future work.

## 6 Conclusions

In this study, we apply a random forest classifier on windowed seismic data to detect and distinguish between noise and seismic signals of slope failures and earthquakes. We test our workflow on two data sets and explore ways to improve its performance on the imbalanced data sets. The improvement techniques allow us to overcome an obstacle that often occurs in natural hazard detection: the deficiency of training data. The advantage of random forest compared to previously suggested approaches using Hidden Markov Models is its simplicity and relative ease of implementation in terms of parameter tuning, post-processing, and 495 computational cost.

We developed a new method to process continuous data streams in near-real-time, which combines detection and identification of rare events. Additionally, this algorithm can outperform a two-step STA/LTA detector and event classifier. The high number of true positives gives us confidence to detect slope failures. However, sub-optimal network configuration, similar frequency content generated by different sources, and low SNR lead to a high false alarm rate. Inclusion of classification results 500 from a second seismic station at regional distance to the study site significantly reduces false positives and is therefore advised for operational use.

We show that two different data sets run into two different types of problems: Low SNR of targeted class and an imbalanced data set. Training data manipulation or small adjustments to the classifier can both mitigate poor classification results due to an imbalanced data set.

Our approach enables us to detect the occurrence of rare events of high interest in a large data set of more than a million windowed seismic signals. Our model, trained on rock slope failures, also detected snow avalanches. It therefore seems that this method is well suited to detect mass movements in general.

Manual review of seismic data is a tedious task, and especially for non-experts, uncertainty and misclassification rates can be high. An automatic classifier, however, can run in the background on a standard machine (in our case 2017, Intel Core i7) 510 and alert stakeholders in case of an event classified as slope failure. Our implementation of a machine learning algorithm for seismogenic mass movement detection may therefore in the future provide valuable support to natural hazard management.

*Code and data availability.* Data supporting this research are available in Geopravent (2017) and are not accessible to the public or research community. To gain access contact Lorenz Meier. Computed feature files and code used in this study are available on https://github.com/michaelawenner/Automatic_classification_Bondo.

 **Appendix A: Slope Failure Signals**

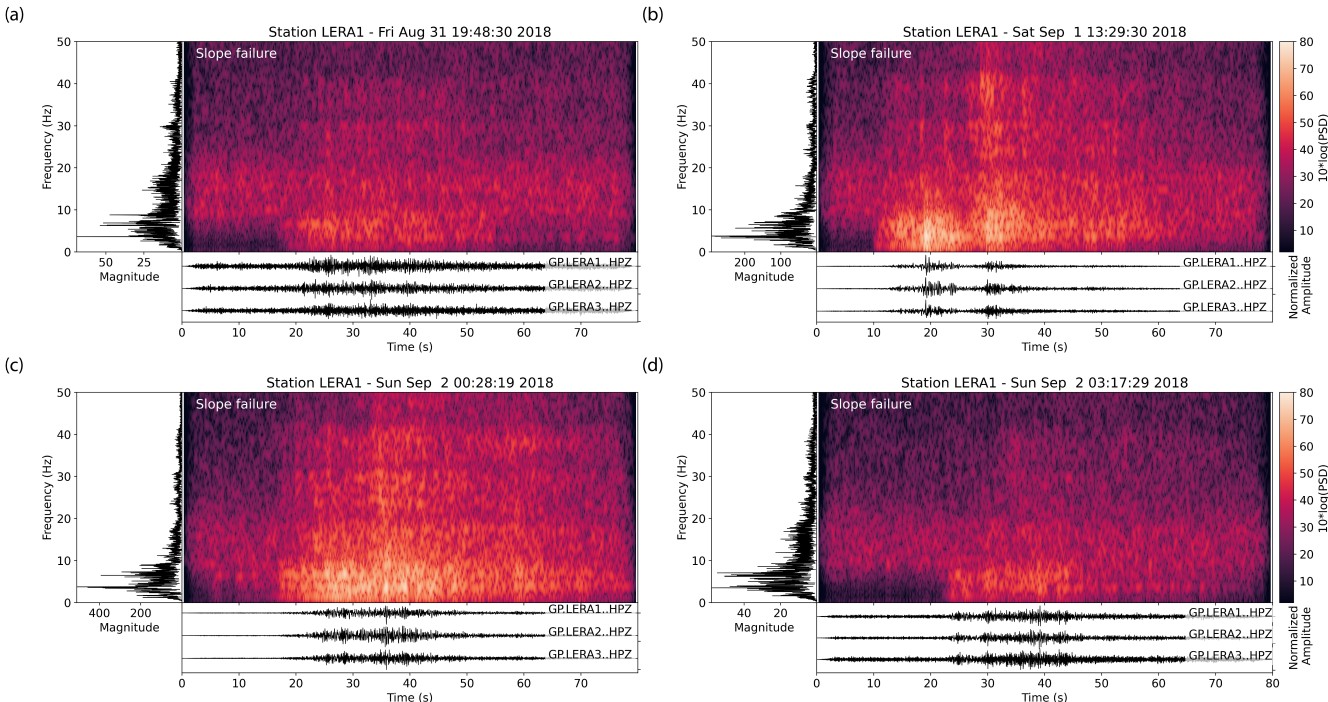

**Figure A1.** Seismic signals, spectrograms and spectra of four additional slope failure events in 2018 used for training. Spectrograms have been computed with a window length of 128 samples, an overlap of 120 samples and an FFT length of 2048.

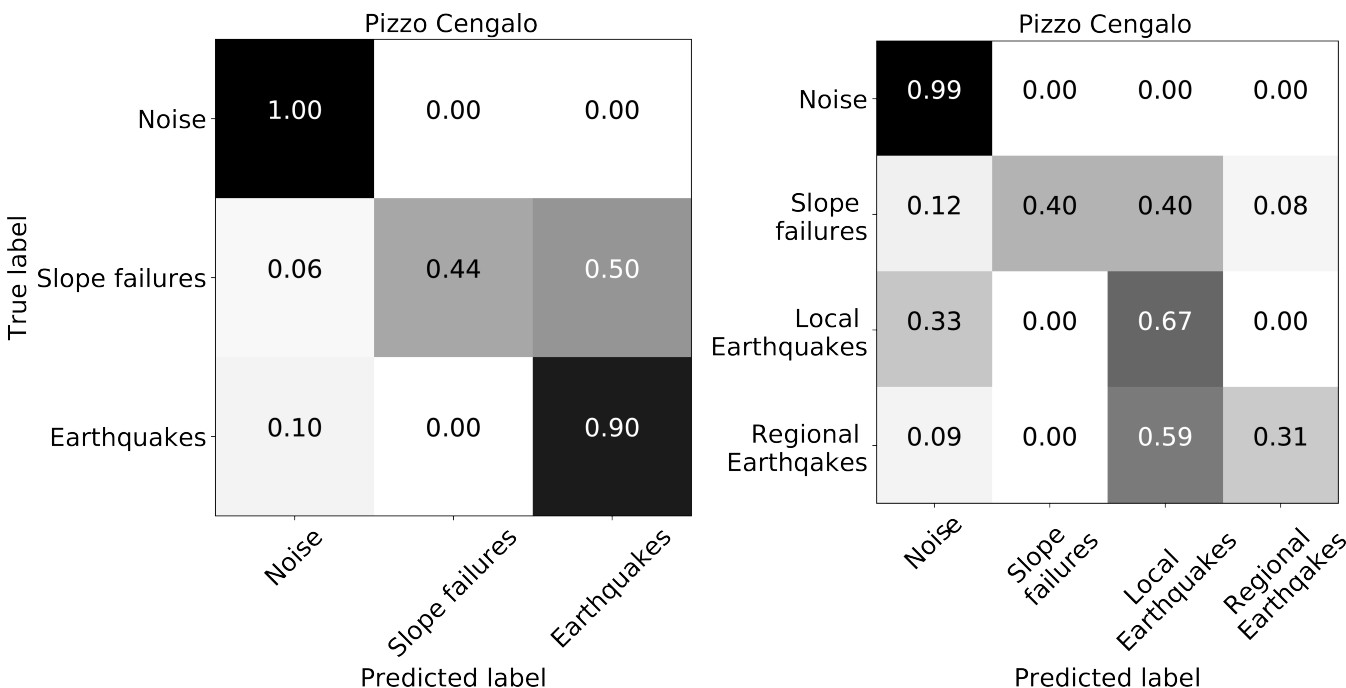

**Figure A2.** Confusion matrix of initial test of random forest on the Pizzo Cengalo data set. Left: all earthquakes lumped into one class. Right: earthquakes divided in local and regional/teleseismic earthquakes.

# Appendix B: List of Computed Features

*Author contributions.* LM with Geopraevent installed the three seismometers and assured data transmission. CH provided the feature computation code. MW processed and analyzed the data with the help of CH and FW. MW prepared the manuscript with contributions from all co-authors.

*Competing interests.* The authors declare that they have no conflict of interest.

*Acknowledgements.* The project is funded by WSL's strategic initiative Climate Change Impacts on Alpine Mass Movements (CCAMM). FW's salary was funded by the Swiss National Science Foundation (GlaHMSeis Project PP00P2_157551 and PP00P2_183719). The canton of Grisons partly payed MW's salary. We thank Geopraevent Ltd., who installed and maintain the seismic stations, provided the data, an online data portal and implemented the algorithm to run in real-time. We are thankful for constructive feedback from Velio Coviello and
an anonymous reviewer on an earlier version of the manuscript. We also want to acknowledge the Obspy developer team Beyreuther et al. (2010); Krischer et al. (2015) for providing an easy to use framework for seismic data handling.

**Table A1.** Table with all features, slightly adjusted from Provost et al. (2017)

| | **Waveform Features:** | |
|---|---|---|
| 1 | Ratio of the mean over the maximum of the envelope | — |
| 2 | Ratio of the median over the maximum of the envelope | — |
| 3 | Kurtosis of the raw signal (peakness of the signal) | $\frac{m_4}{\sigma^4}$, with $m_4$: fourth moment, $\sigma$: standard deviation |
| 4 | Kurtosis of the envelope | see 3 |
| 5 | Skewness of the raw signal | $\frac{m_3}{\sigma^3}$, with $m_3$: third moment |
| 6 | Skewness of the envelope | see 5 |
| 7 | Number of peaks in the autocorrelation function | — |
| 8 | Energy in the first third part of the autocorrelation function | $\int_0^{T/3} C(\tau)d\tau$, with $T$: signal duration, C: autocorrelation function |
| 9 | Energy in the remaining part of the autocorrelation function | see 8 |
| 10 | Ratio of 8 and 9 | — |
| 11 – 15 | Energy of signal filtered in $1 — 3\,\mathrm{Hz}$, $3 — 6\,\mathrm{Hz}$, $5 — 7\,\mathrm{Hz}$, $6 — 9\,\mathrm{Hz}$ and $8 — 10\,\mathrm{Hz}$ | $\int_0^T y_f(t)dt$, with $y_f$: filtered signal in the frequency range [f1-f2] |
| 16 – 20 | Kurtosis of the signal in $1 — 3\,\mathrm{Hz}$, $3 — 6\,\mathrm{Hz}$, $5 — 7\,\mathrm{Hz}$, $6 — 9\,\mathrm{Hz}$ and $8 — 10\,\mathrm{Hz}$ frequency range | see 3 |
| 21 | Maximum of the envelope | — |
| | **Spectral Features:** | |
| 22 | Mean of the DFT | DFT: discrete Fourier transform |
| 23 | Max of the DFT | — |
| 24 | Frequency at the maximum | — |
| 25 | Frequency of spectrum centroid | — |
| 26 | Central frequency of the 1st quartile | — |
| 27 | Central frequency of the 2nd quartile | — |
| 28 | Median of the normalized DFT | — |
| 29 | Variance of the normalized DFT | — |
| 30 | Number of peaks ($> 0.75$ DFTmax) | $\mathrm{DFT}_{\max}$: maximum of the DFT |
| 31 | Mean value for the peaks | — |
| 32 – 35 | Energy in $[0, \frac{1}{4}]\mathrm{Nyf}$, $[\frac{1}{4}, \frac{1}{2}]\mathrm{Nyf}$, $[\frac{1}{2}, \frac{3}{4}]\mathrm{Nyf}$, $[\frac{3}{4}, 1]\mathrm{Nyf}$ | $\int_{f_1}^{f_2} \mathrm{DFT}(f)df$ with $f_1$, $f_2$: the considered frequency range |
| 36 | Spectral centroid | $\gamma_1 = \frac{m_2}{m_1}$, with $m_1$ and $m_2$ the first and second moment |
| 37 | Gyration radius | $\gamma_3 = \sqrt{\frac{m_3}{m_2}}$, with $m_3$ the third moment |
| 38 | Spectral centroid width | $\sqrt{\gamma_1^2 - \gamma_2^2}$ |

**Spectrogram Features:**

| # | Feature | Formula |
|---|---------|---------|
| 39 | Kurtosis of the maximum of all discrete Fourier transforms (DFTs) Kurtosis as a function of time t | $\text{Kurtosis}\left[\max\limits_{t=0,\dots,T}\left(\text{SPEC}(t,f)\right)\right]$ with SPEC(t,f): spectrogram |
| 40 | Kurtosis of the maximum of all DFTs as a function of time t | see 39 |
| 41 | Mean ratio between the maximum and the mean of all DFTs | $\text{mean}\left(\frac{\max(\text{SPEC})}{\text{mean}(\text{SPEC})}\right)$ |
| 42 | Mean ratio between the maximum and the median of all DFTs | see 41 |
| 43 | Number of peaks in the curve showing the temporal evolution of the DFTs maximum | — |
| 44 | Number of peaks in the curve showing the temporal evolution of the DFTs mean | — |
| 45 | Number of peaks in the curve showing the temporal evolution of the DFTs median | — |
| 46 | Ratio between 43 and 44 | — |
| 47 | Ratio between 43 and 45 | — |
| 48 | Number of peaks in the curve of the temporal evolution of the DFTs central frequency | — |
| 49 | Number of peaks in the curve of the temporal evolution of the DFTs maximum frequency | — |
| 50 | Ratio between 48 and 49 | — |
| 51 | Mean distance between the curves of the temporal evolution of the DFTs maximum frequency and mean frequency | — |
| 52 | Mean distance between the curves of the temporal evolution of the DFTs maximum frequency and median frequency | — |
| 53 | Mean distance between the 1st quartile and the median of all DFTs as a function of time | — |
| 54 | Mean distance between the 3rd quartile and the median of all DFTs as a function of time | — |
| 55 | Mean distance between the 3rd quartile and the 1st quartile of all DFTs as a function of time | — |

**Table B1.** Number of events and number of windowed seismic data (20s windows as an example) of the training and test data set of Pizzo Cengalo and Illgraben

|  |  | Events in data set | | Number of 20s windows | |
|---|---|---|---|---|---|
|  |  | Pizzo Cengalo | Illgraben | Pizzo Cengalo | Illgraben |
| Training data | Noise | 1025 | 222 | 2095 | 473 |
|  | Slope failures | 3 | 59 | 76 | 337 |
|  | Earthquakes | 17 | 66 | 438 | 674 |
|  |  |  |  |  |  |
| Test data | Noise | 438 | 85 | 836 | 160 |
|  | Slope failures | 2 | 31 | 54 | 170 |
|  | Earthquakes | 13 | 34 | 373 | 373 |

**Table B2.** Random forest parameters

| | |
|---|---|
| Number of trees | 2000 |
| Split quality measure | Gini criterion |
| Minimum number of samples required to be a leaf node | 4 |
| Maximum depth of a tree | 60 |
| Minimum number of samples for an internal node to be split | 2 |

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

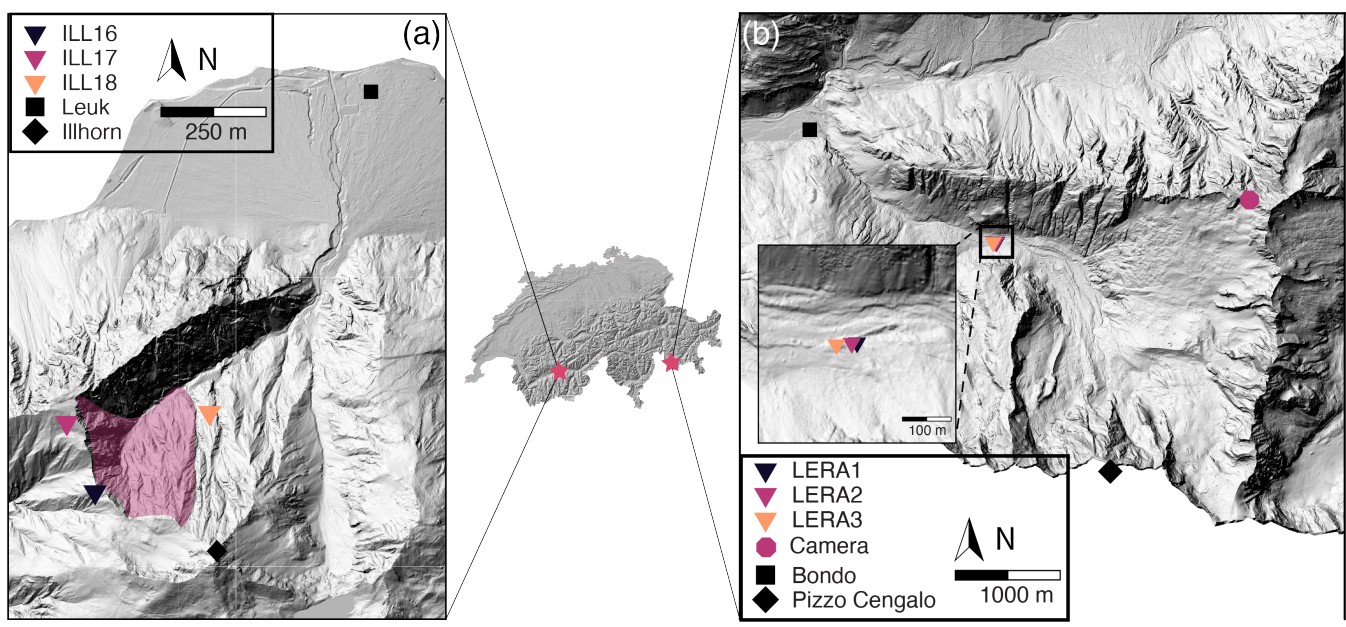

**Figure 1.** Illgraben catchment (left) and Bondasca valley (right) with locations in Switzerland (stars in middle panel).(a) Overview of the Illgraben catchment in Switzerland and locations of stations ILL16 – 18 depicted as colored triangles. (b) Overview of the Bondasca valley. Location of LERA network depicted as colored triangles. Zoom-in shows individual seismic stations (LERA1 – 3) in the upper right corner.

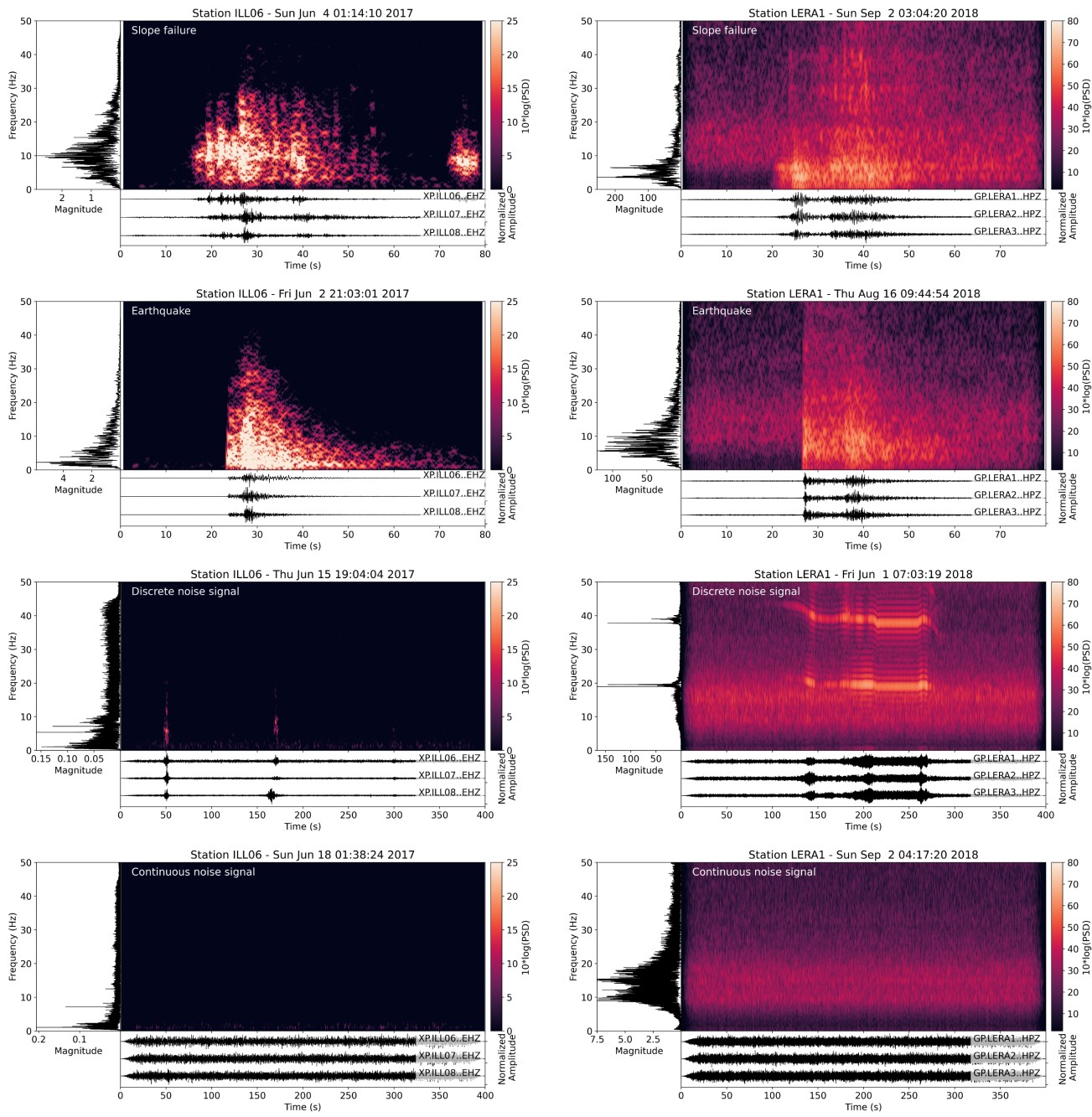

**Figure 2.** Waveforms, spectrogram and spectra of example events for each class recorded at Illgraben (left) and Pizzo Cengalo (right)

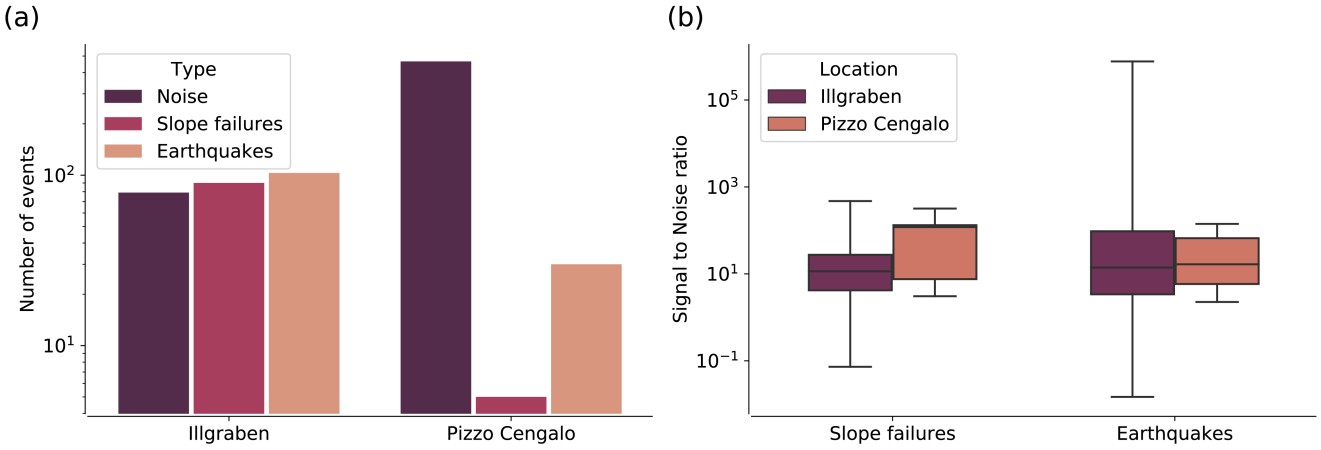

**Figure 3.** (a) Bar chart with events per class for both the Illgraben and the Pizzo Cengalo data set. (b) Boxplot of SNRs of all slope failure and earthquake events at Illgraben and Pizzo Cengalo. The boxes denote the quartiles (25 - 75%), the black horizontal line denotes the mean and the black vertical error bars indicate outliers.

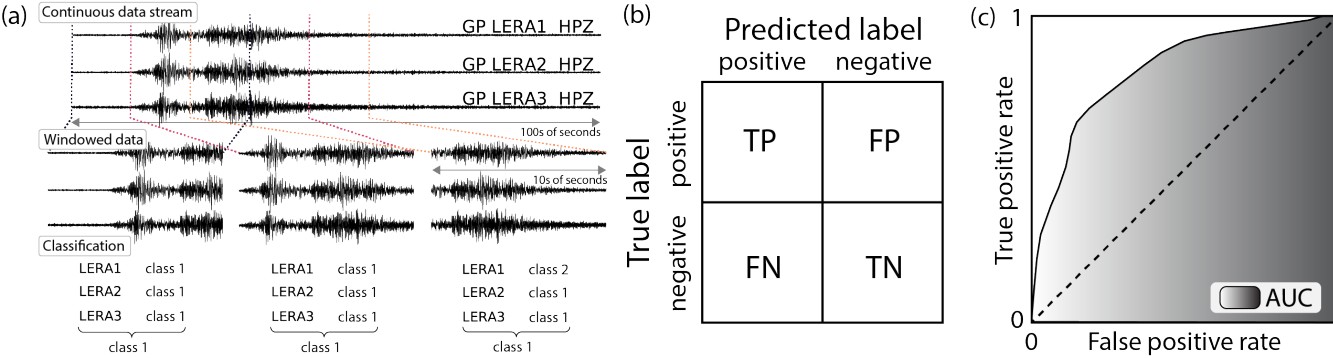

**Figure 4.** (a) One-step classification scheme with continuous data stream, windowed data, classification per station and label for window, (b) Confusion matrix for a two class problem (positive, negative), with true labels as rows and predicted labels as columns. True positives (TP) and true negatives (TN) on the diagonal and false negatives (FN) and false positives (FP) on the off diagonal elements. (c) ROC curve with true positive rate (TPR) on the y axis and false positive rate (FPR) on the x axis. Shaded area (AUC) measures model accuracy.

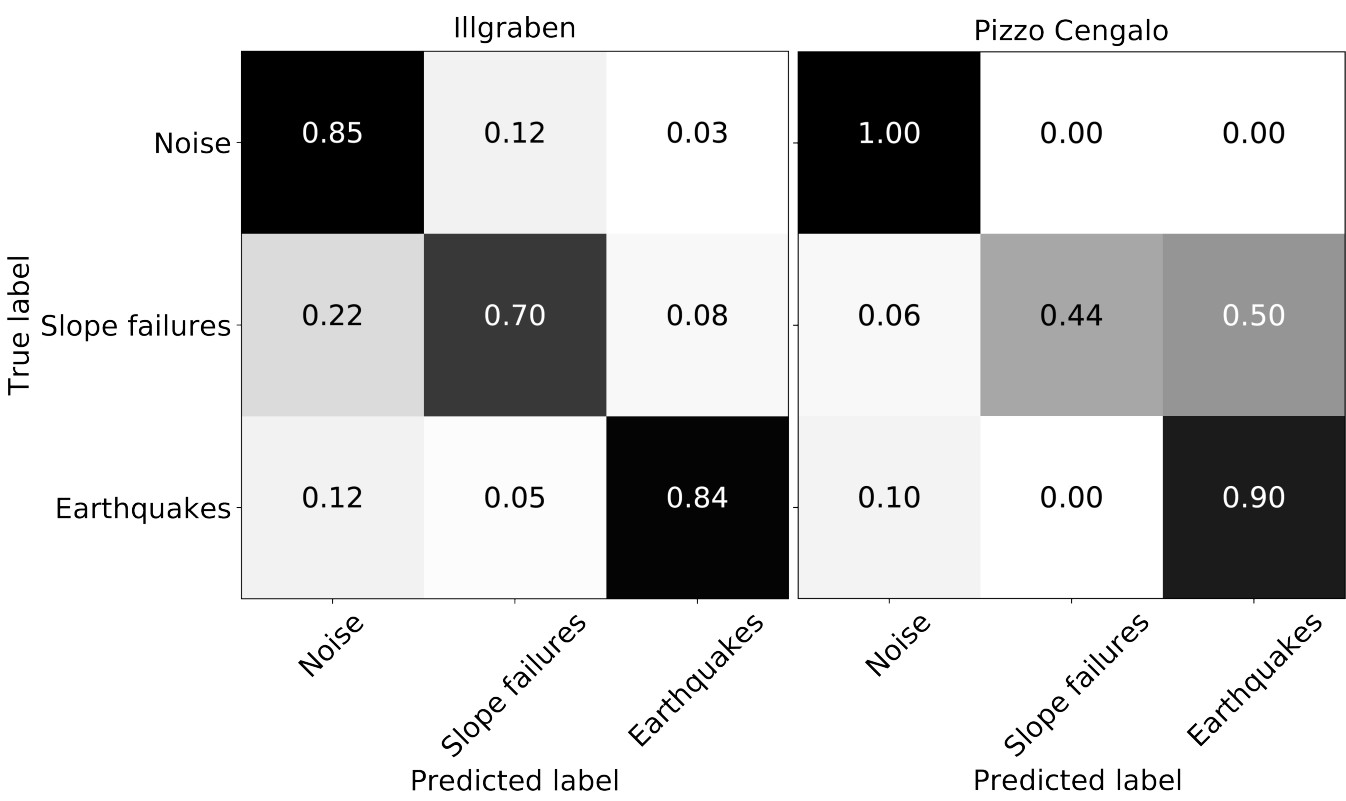

**Figure 5.** Confusion matrices of an initial test of the continuous random forest approach on 20 s windowed data. Left: Illgraben data set, right: Pizzo Cengalo data set.

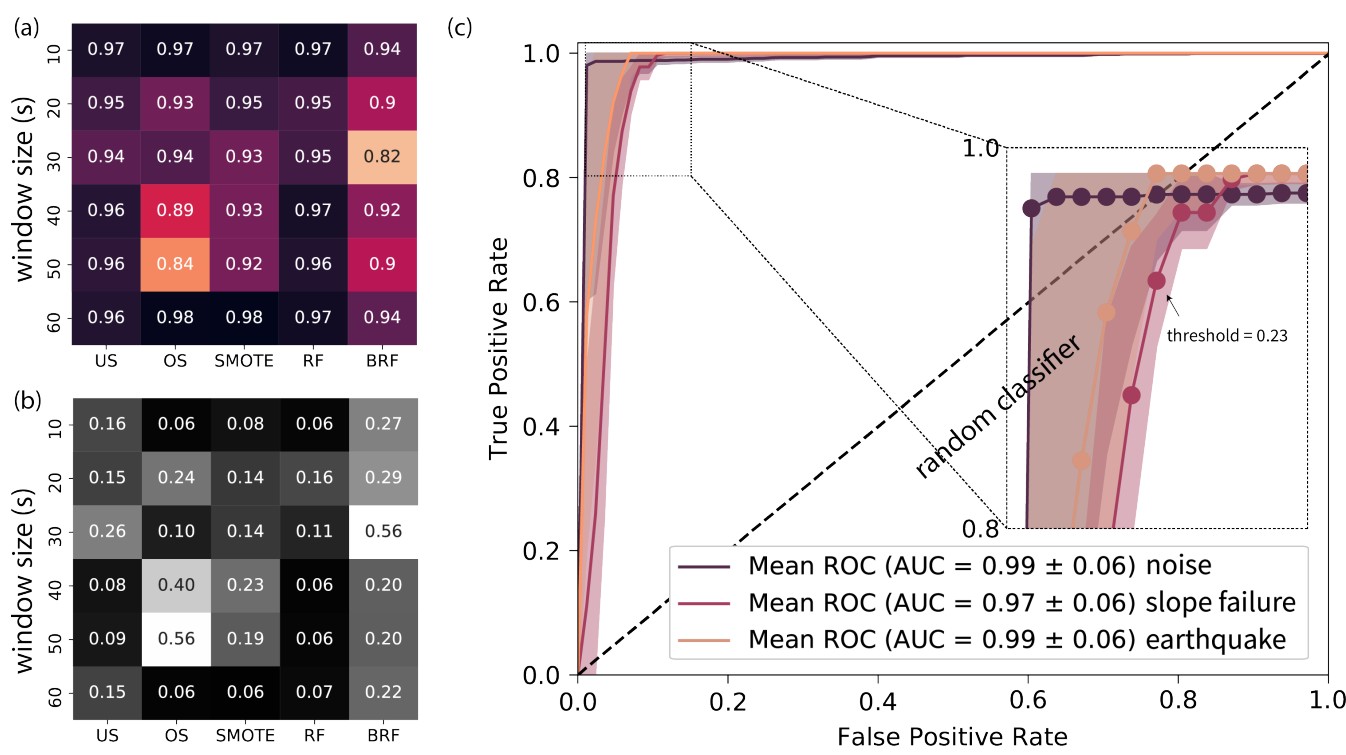

**Figure 6.** ROC analysis on the Pizzo Cengalo data set. (a) Heatmap of AUC values for SF class for different window sizes and different methods to handle imbalanced data sets, namely undersampling (US), oversampling (OS), synthetic minority oversampling (SMOTE), random forest with original data set (RF), balanced random forest (BRF). The darkest colors denote the largest AUC values. (b) Heatmap of 95% confidence interval of AUC values. Darker colors denote smaller standard deviation (c) ROC curve for 40 seconds time window and random forest. 3-fold cross validation with mean as solid line and 95% confidence interval. Zoom into corner with circles illustrates TPR and FPR values for different thresholds. Probability threshold for TPR rate of SF class > 0.9 is depicted.

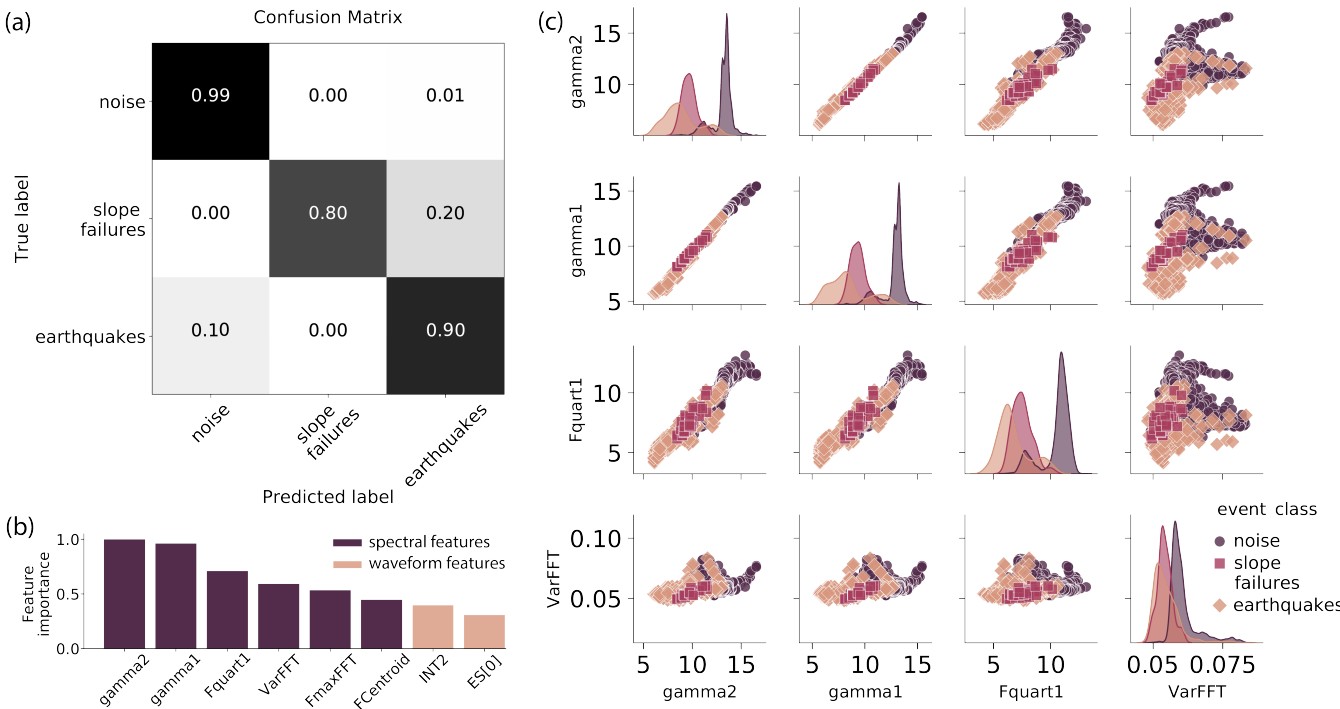

**Figure 7.** Confusion matrix and feature importance analysis of the optimized classifier for the Pizzo Cengalo data set. (a) Normalized confusion matrix of final model test. The darker the colors, the higher the values. For an ideal classifier, all samples would be located on the diagonal. (b) Eight most distinct features normalized to one. Dark columns mark spectral features (characteristics of signal in frequency domain), light columns mark waveform features (characteristics of signal in time domain). Labels: spectral gyration radius (gamma2), spectral centroid (gamma1), central frequency of the first quartile (Fquart1), variance of the normalized FFT (VarFFT), frequency at the maximum of the FFT (FmaxFFT), frequency at spectrum centroid (FCentroid), energy of the last 2/3 of the autocorrelation function (INT2), and the energy of the seismic signal in the frequency band of 1-3 Hz (ES[0]) Provost et al. (2017). (c) Pairplot of four most distinct features. Per cell two features plotted against each other, except for diagonal. Diagonal shows univariate distribution of the feature. Colors mark different event classes

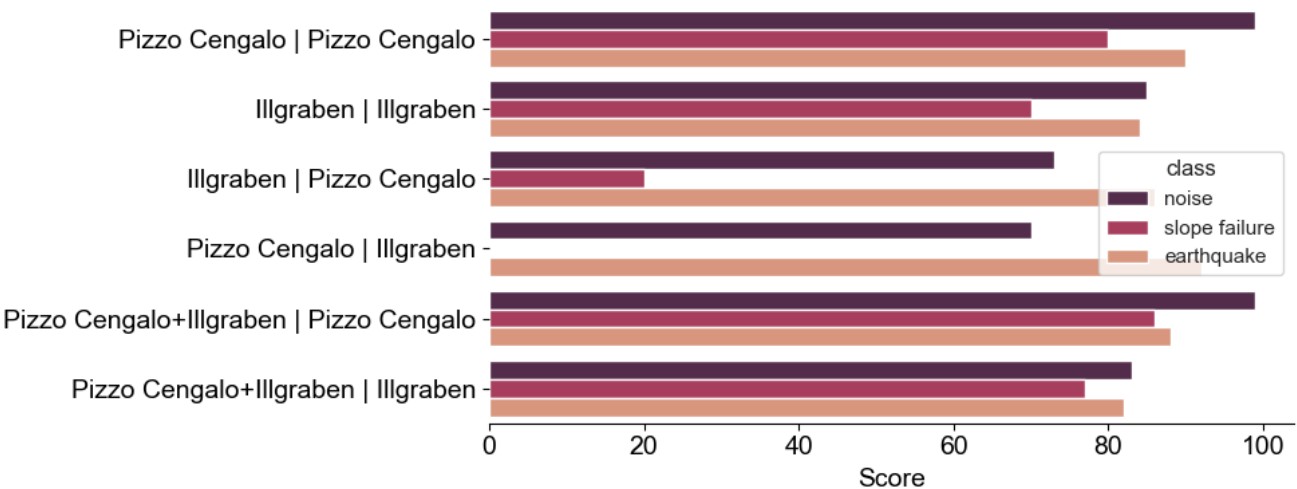

**Figure 8.** Automatic camera photos of slope failures in 2019. Photos from April show the avalanche deposits from the events classified as slope failures of 4 April and 26 April. Photos from July and August show photos before and after the events of 16 July and 14 August. For the 14 August event, no deposit is visible on the photos.

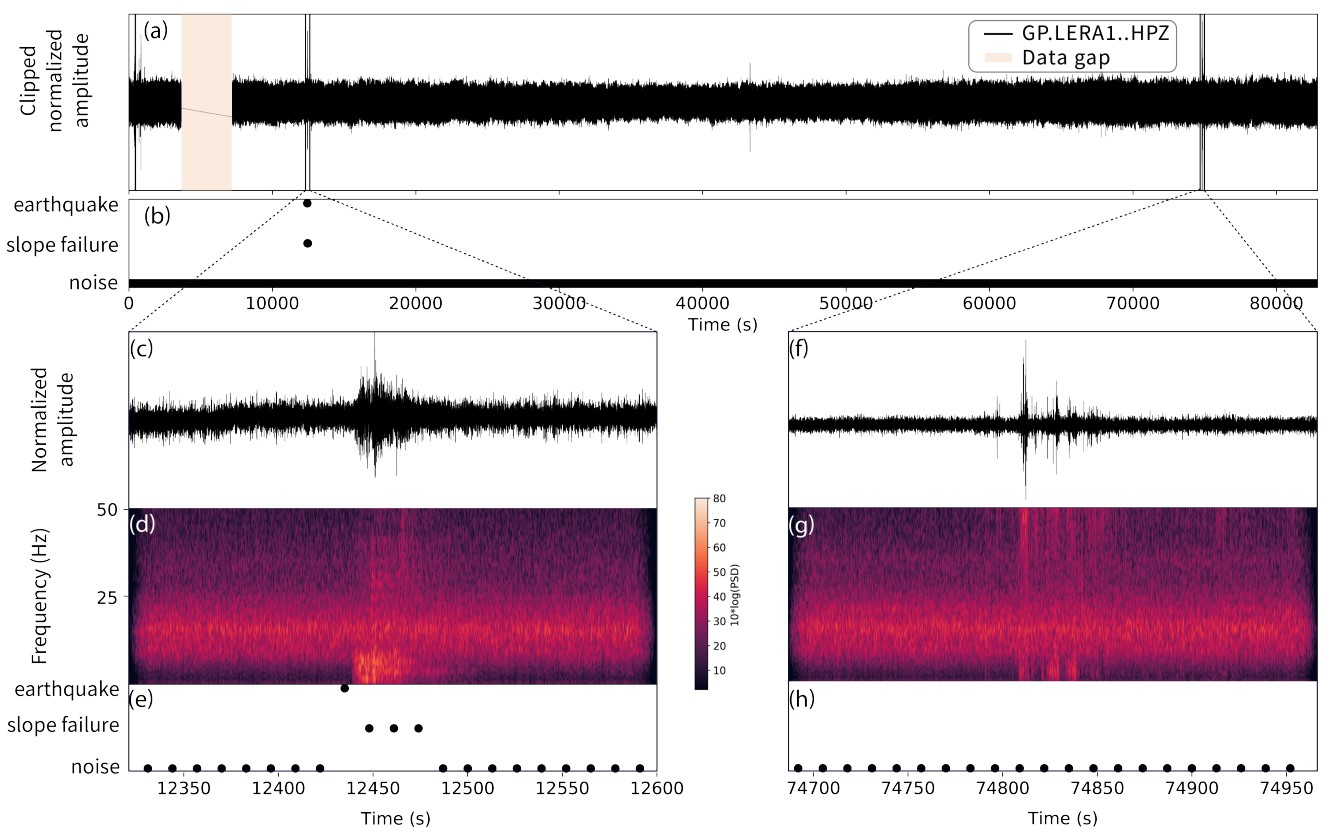

**Figure 9.** Classifier tested on one day of data from Pizzo Cengalo in 2019 (July 16). (a) Waveform of one day, with data gap (orange area) (b) label of each 40 seconds time window. (c) Waveform and (d) spectrogram with spectral power of a slope failure and (e) associated classifications. (f) Noise event with (g) spectrogram and (h) associated classifications.

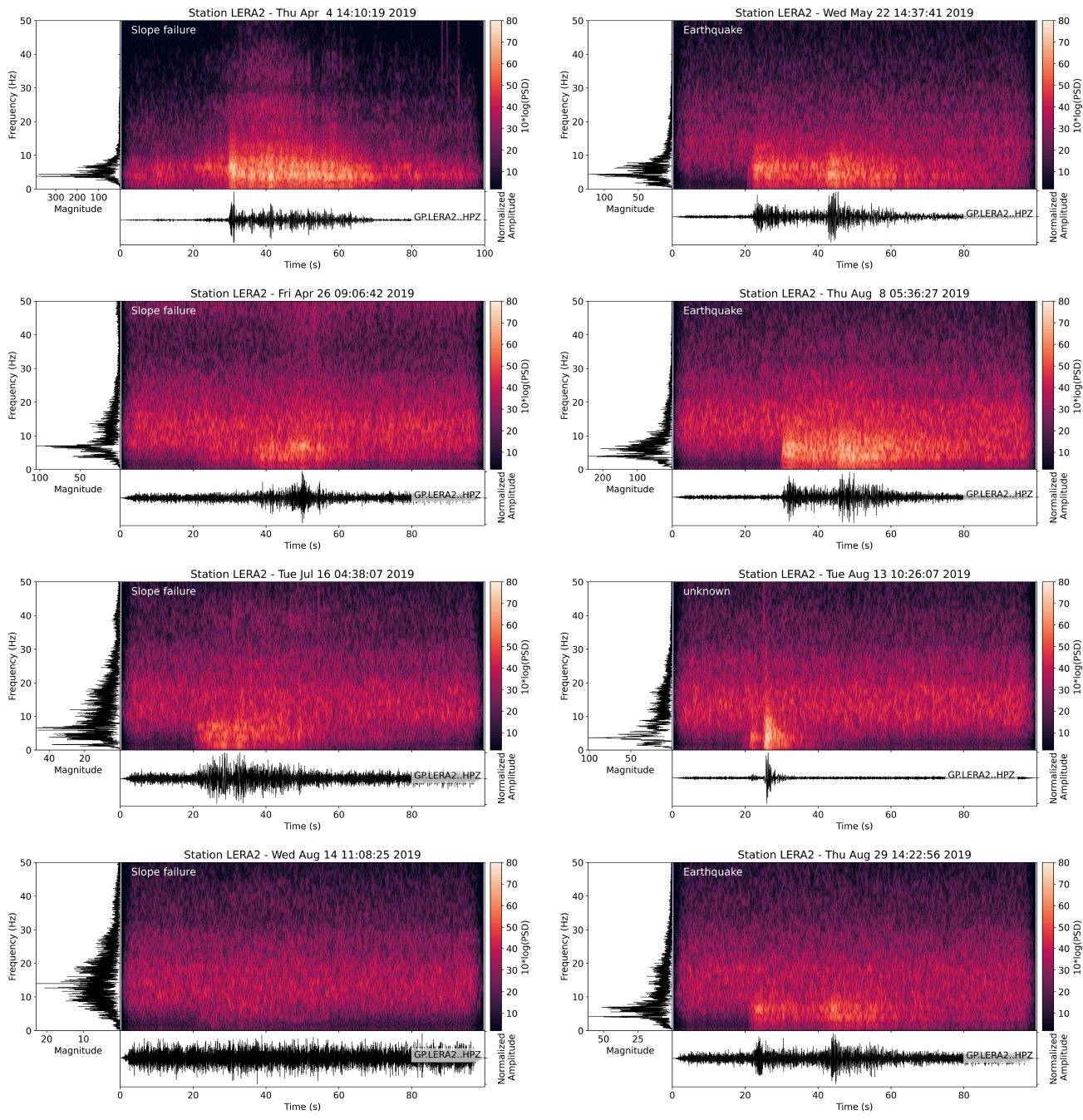

**Figure 10.** Waveforms, spectrograms and spectra of all events classified as slope failure in 2019 at Pizzo Cengalo, as well as the missed slope failure on August 14, 2019 (bottom left). White labels on spectrograms mark the type of event.

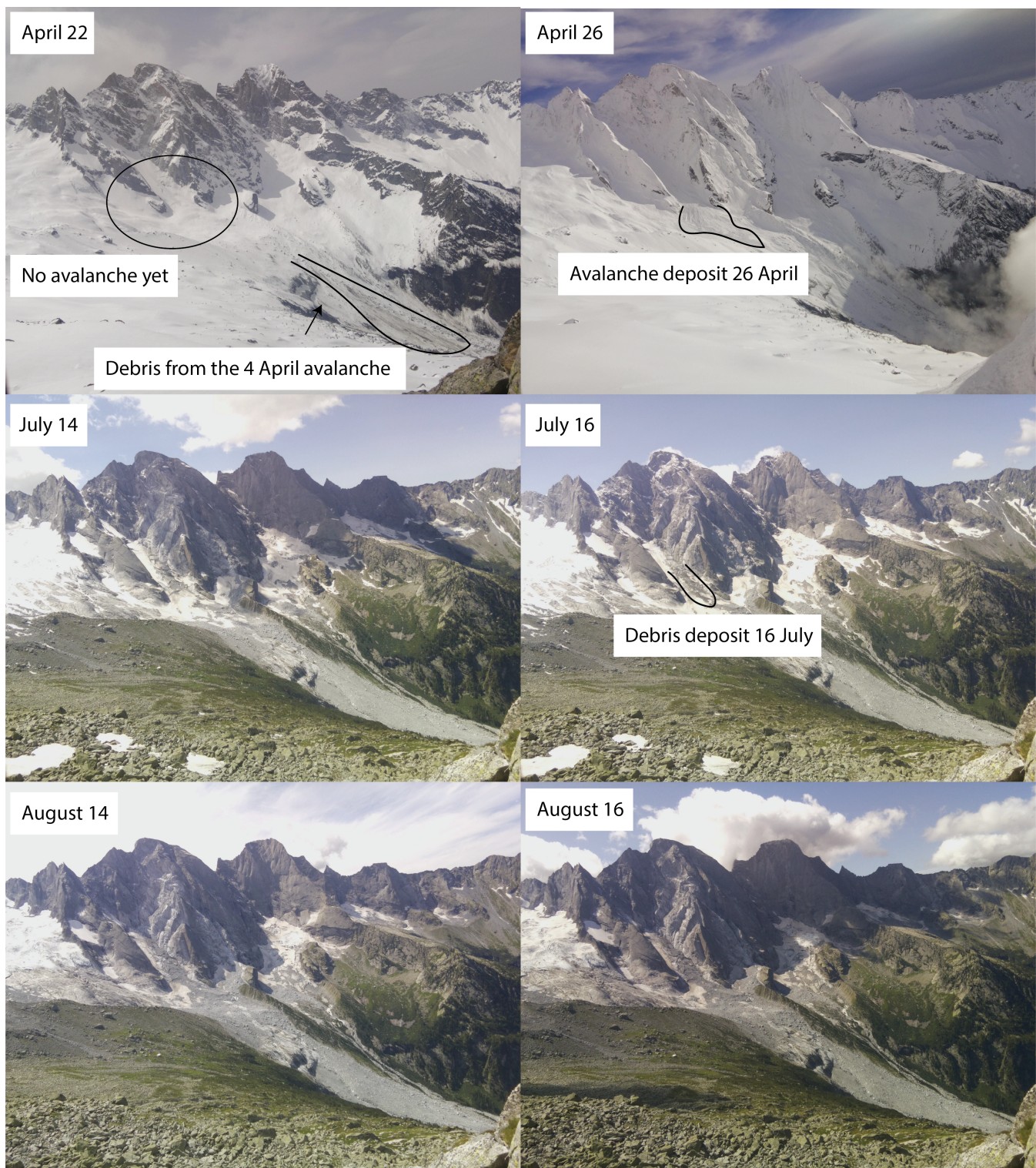

**Figure 11.** Scores for each class with different training and testing data sets. Labels on the y-axis show the data set(s) used for training before the vertical bar, and the data set used for testing after the vertical bar.