# Peer review of "Near Real-Time Automated Classification of Seismic Signals of Slope Failures with Continuous Random Forests"

_Natural Hazards and Earth System Sciences, 2020_

## Referee Comment (RC1) · Anonymous Referee #1 · 27 Jul 2020

I read a study that has explored the potential of a machine learning algorithm to jointly detect and classify mass wasting and earthquake events from a small linear geophone array along a channel in the Swiss Alps. The study opens a new and timely avenue of "close to real time" hazard event warning by combining state of the art approaches in an arguably not optimally suited experimental setup. It discusses these drawbacks as well as different ways to account for them. The document is mostly well structured, provides adequate background, justification and motivation of the study. The applied/developed methodology is clearly described and can be digested without major ambiguities. The study is well placed in the scope of the journal and I am confident that after some modifications, it will be a valuable addition to the journal's portfolio.

[Figure]

Indeed, as the authors point out, the study is faced with suboptimal boundary conditions. The most important drawbacks are i) network geometry (linear array with 20 m station spacing), ii) a lack of independent control on the hillslope events and iii), a striking event type imbalance ($10^1$ hillslope events, $10^2$ earthquakes, $10^3$ noise cases). All these drawbacks are transparently mentioned, and their impact and counter measures are discussed in the text. Consequently, from a technical perspective, there is no reason to worry. However, it strikes my why this study design has been chosen to work with from the beginning. Why has this timely, rigorous and relevant study not been set up at a more suitable study site? There are many examples (cited in the text) where the network geometry is better (perhaps even in including a section of linear and densely spaced sensors to test the impact of such conditions, e.g. at the Sechilienne landslide), where there is excellent independent control on location, magnitude and to some degree the timing of hillslope activity, and where overall there are significantly more hillslope failure events that would lead to a less imbalanced data set? Somewhat, this excellent idea and study approach is vastly undersold due to the quality of the data. Currently, a wider impact is impeded by the big question marks on the representativeness given that only a handful of slope failure events has been detected and this with a 270 % error (3 seismogram interpreted hillslope events versus 8 random forest-based hillslope events). Regarding the latter, while the abstract sounds quite confident (80 % prediction accuracy), the implementation of the approach does not. And it is a bit contradictory to claim the random forest approach would overcome manual inspection efforts to correctly classify an event, whereas in the discussion it becomes necessary to judge manually, which of the eight detected events is due to hillslope activity and which is an earthquake. Long story short, I see two points that should receive more attention in the manuscript: i) a robust justification of the study site and experiment setup (Why working with an obviously unsuited network and missing event control?), and ii) a more thorough discussion of the classification errors, with due respect to the very small number of actual events and the resulting implications for the overall uncertainty.

Regarding the classification quality part, one way that might be worth to explore is

to use the hillslope events from the entire data set, not just the training subset. This of course only in the exploration of the classification quality (sections 4.1 and 4.2). The idea is to reduce the imbalance by increasing the number of hillslope events. In addition, this would shed some light on the actual impact of 5 versus 8 hillslope events.

The results section partly grades into a discussion. I recommend keeping these things separated, especially since there is a dedicated discussion section. Examples are l221-224, l249-254, l261-262, l268-269, l271-272.

l20, I do not think it is necessary to use climate change as driver of this study. As in the abstract, it is sufficient to motivate by the mass movements, alone. But this is just a recommendation. No need to stick to that.

l34-35, check journal guidelines about order of references, here and throughout. Commonly, this is by date or author name, rather than apparently random order.

l39-40, the larger amplitudes of slope failures must be compared to something. I assume you mean tremors. But the distance to the source will dominate the amplitude discussion. I suggest, to remove this misleading part of the sentence, it is of limited use, here. Overall, I am not sure the comparison of rock avalanches to tremors is a good one, especially in this journal and its readership.

l48-54, well summarised. I suggest to pick that up in the discussion again, because like your routine the HMM approach also generates near-real time classification of events. Thus, a verbal comparison of pros and cons of the two approaches is something the reader is interested in, and for good reason. Ideally, one would benchmark both approaches using the same input data, but I fear this is not feasible, here.

l 55, the section about STA/LTA picking is a bit unfortunate, here. In the above paragraph you discuss detecting and classifying. Here you go back to just detecting. Would it not be more intuitive to first give a general introduction that defines and distinguishes detection and classification, and then elaborates on the different approaches to these

two tasks? I suggest to write such a short introduction prior to l. 48. Then you can list the different approaches.

l76-77, that last sentence of the paragraph is actually results and discussion. I recommend to remove it here.

l78-80, in your scope, points a) and b) are not actually discussed and investigated. You do not write about decreased slope activity as a precursor of larger events or transitions of hillslope to channel activity. In fact, you cannot do this with only a hand full of events in total. I suggest to reword these points, here. Or simply collapse this paragraph with the above one after the corrections have been implemented.

l86, check SI unit conformity of volume numbers. Also see journal guidelines.

l92, you may want to add more information about the loggers and recording frequencies, as well as on the installation of the senors (surface, depth, coupling)?

l98, in the methods, I recommend adding the benchmark efforts that you discuss in section 5.2. This is a laudable and insightful test and it must be justified and described in the methods section.

l101, check conformity of closing parenthesis in figure reference. Also, in other parts of the manuscript, this parenthesis is missing, check for correctness and consistency.

l134-135, this sentence kind of glances over a maybe important topic. Is there any way to show this more rigorously? I might suspect that i) local versus teleseismic earthquakes are quite distinct in terms of labeled features and ii) that smaller local quakes might be more similar to slope activity. Thus, could this lumping not be one reason for the result of 5 out of 8 hillslope event classifications being earthquakes? Usually, sentences that start with "After rigorous testing..." tend to hide potentially important subjective decisions instead of transparently showing the foundations of these decisions. Consequently, it would be good to be more transparent here, and show the effect of the lumped case versus for example two or three earthquake classes. Or at

least to discuss why for random forests it may be appropriate to stick to very small numbers of classes.

l187, to account for the bias due to the imbalanced data set, can you not calculate the confusion matrix based on log-scaled numbers? I think in one of the Hammer HMM papers this has been done.

l216-217, why different colour schemes for the two matrices? It is not intuitive. No big deal but I may mention that it took me some thought to wonder why these different colours. Unless there is a reason (which should then be mentioned in the text/caption) I suggest to use the same colour scheme.

l220, reword, currently it reads as if RF and BRF are techniques at the same level as RF with US, OS and SMOTE. From the methods I read that US, OS and SMOTE are data manipulation steps prior to a subsequent RF classification, no? Also, it would be good to actually discuss these findings later on (section 5). What does it mean that the imbalance countermeasures do not yield any improvement, but rather decrease the quality of the classification? What can we learn from that? What might be the reason?

l230, this number of 2 RF in the test data set comes out of the blue. Please revise and mention this at an appropriate place.

l258, the manual classification parameters must be defined in the methods (What are your classification judgements based on?). The image and radar methodology must be mentioned, as well. Also, since the catalogue is a key feature to validate your approach, I recommend to spend significantly more than just one short sentence on this topic, both in the methods description and the presentation of the resulting catalogue (a table or in the text).

l262-263, I suggest you give more details here, in terms of description of the events. It is only three failures, so there is space for that and it is important as the main goal of your study is to work out such events. Based on which criteria did you define these

signals as hillslope failures? What are the event's properties? Also, in fig. 6, I only see one event and not all three. I suggest to plot the PSDs and seismograms also for the two other events, as in fig.6 c-e-g.

l268-269, this is an unsupported statement. How are we to judge that this was an earthquake without seeing any data of it? Why do you think it is no hillslope event? Please present a PSD and seismogram as well as a more detailed description of the properties. This is the results section and it should present results sufficiently clear and exhaustive to allow you to draw conclusions from it.

l302, can you quantify this statement? What means high SNR, compared to what?

l310, as mentioned above, this section should be motivated and described in the methods section, already. And it's outcomes should be described in the results section, so that you can focus on the implications, here. Please revise.

l311, delete comma after "shown".

l333, this is a valuable finding but strikingly out of context. Either include the runoff classification part from the beginning or leave it out (I recommend the latter). Also, runoff appears to be a continuous feature rather than a comparably short lived event. In fact all PSDs of the manuscript show the seismic signature of water runoff. So why classifying it and how handling the case of two "events" occurring at the same time, such as runoff and rockfall?

l342, revise this first sentence. Yes it is feasible, but with an error of 230 % (3 times right, 5 times wrong).

l345, rewrite "is a challenge that an imbalanced training data set enhances". Do you mean a challenge that is due to an imbalanced training data set? Or a challenge that may be solved by a less imbalanced training data set?

l349, manual inspection is not just advisable but crucial to account for the issue of misclassification, see comment two above. In the same line, replace "then" by "than"

and "monitoring" by "inspecting".

l350-353, these are arm waving sentences. Either expand on this topic or leave it out. Currently this does not help the reader much. What is behind semi- and unsupervised ML algorithms, more specifically? Which specific drawbacks of the current approach would they solve? What are "unseen patterns"? I summary, I suggest not to mention this part, unless you find a way to explain its value in more detail.

I did not check the references for consistency and correct formatting.

Fig2c, value of that sketch is very limited. You may consider removing this panel.

Fig.3, check font sizes, this is a really small font, hard to read. See journal guidelines on minimum size.

Fig 4, a and b homogenise colour schemes.

Fig 6, as mentioned above, also show other hillslope events, as well. Font on legend colour bar is too small.

---

## Referee Comment (RC2) · Anonymous Referee #2 · 10 Aug 2020

**Near Real-Time Automated Classification of Seismic Signals of Slope Failures with Continuous Random Forests**

*Submitted to NHESS*

M. Wenner, C. Hibert, L. Meier and F. Walter

August 9, 2020

**General comments**

This paper presents the training, testing and application of a near-real time automated classifier of seismic signals. The text reports whether accurate identification of slope failure (SF) events is feasible using consecutive time windows on relatively noisy data from a small seismic network with little aperture. These are sub-optimal conditions for such an analysis, but so is often the case in areas with irregular topography and difficult access. Building on previous studies that relied on denser seismic networks and higher signal to noise ratios (SNR), the authors first train the algorithm with a dataset containing 1032 noise events, 3 SF and 17 earthquake signals. Then, the algorithm is tested on 441 noise events, 2 SF and 13 earthquake signals, using a running window with overlap on the continuous data stream; a prediction accuracy of 80% is achieved for SF. Finally, the algorithm is applied to 2019 data; three out of four actual SF are correctly identified, but the algorithm also misidentifies five earthquakes as SF. The authors conclude that the presented method can be applied to near-real time continuous monitoring, and outline that this type of algorithms can substantially decrease operator time devoted to manual inspection of seismic data. This topic is well within the scope of NHESS and can be of interest to the communities of environmental seismology, natural hazards, early warning, and beyond.

Strong points of this study are (1) the introduction of a novel application of a random forest classifier using continuous data. The data stream is challenging, similar to what can be expected in hazardous, but increasingly transited and populated, high mountain settings; (2) performance of the presented algorithm is compared to that of a 2-step, well-established algorithm: A STA/LTA detector and classification using the full, i.e. STA/LTA-bounded, event. It is shown that, on the 2019 data, performance of the presented algorithm is superior to that of the 2-step algorithm; and (3) in this initial version, the text reads reasonably well and is, in general, adequately complemented by good-quality figures. Finally, the authors deserve credit for making publicly available the code used in this analysis.

In its current form, this paper suffers from (1) insufficient discussion of several important points, including the dataset, detection choices, misclassification of events, and advantages and potential issues of the chosen method. The authors are using a dataset with very few SF events to train, test and apply the algorithm, in addition to a low SNR of the application data. This results in low SF classification accuracy for the 2019 application data (3/8), so a careful and thorough discussion of these important points should be provided; and (2) several confusing points along the manuscript, due to unclear writing, insufficient explanation or both, which hinder comprehension of the choices and interpretations made. Therefore, before publication can be granted, I recommend a thoughtful review of the aspects detailed below.

**Specific comments**

(Listed in approximate order of importance)

What is the goal of this study? Is the goal to detect slope failures (SFs) with high accuracy even when using running windows on a few number of relatively noisy channels? Is it to test whether any potential SF can be detected, to minimize operator time devoted to visually inspecting seismic data? Is it to test how much accuracy can be achieved with this algorithm applied to realistic noisy data and how it compares to the combination of a STA/LTA-type detector and a random forest classifier on the full event signal? The goal of the analysis should be clearly stated, not implicit in the text, since this is key for reading (and evaluating) the study. In line 78, the authors mention that the scope (do they mean goal?) of this study is to present an algorithm capable of the following "*a) detect an increase in slope activity as an early warning for a plausible larger event in the near future and b) detect rock slope failures that possibly transition into hazardous debris flows early on, to enable down-stream communities to take action*". However, the implications of a) and b) no longer appear in the text. Is an algorithm capable of a) always capable of b), and viceversa? Are there differences between signals of slope-related seismic activity before larger SF, and of SF that represent the beginning of debris flows? Is the trained algorithm capable of both a) and b)? If this is stated as the goal, then it should be discussed later. If this is just the general frame of the study, and the authors are merely naming two hazards that potentially generate precursor SF that can be detected, then the goal should be clearly listed and it should be ensured that there is no confusion.

The instrumentation and dataset are very briefly presented in sect. 2 and 3. A careful explanation of the reasons for their choosing is required, because the network and dataset should be adequate for the study goals. Hence, the key issue here is to clearly justify that this network is adequate for this study. In the introduction, line 72-74, it is mentioned that "*Previous local-scale approaches have used networks designed by experts and set-up as an array, ideal for monitoring such processes. However, due to cost and time constraints, this is not possible and not the case for most potential hazard sites*". References should be added. Based on this, the authors state next that "*We show that by adjusting our methodology to work with a network of low-cost seismometers with a sub-optimal network configuration, the detection of slope failures is still possible without post-processing*". However, the following are still not clear: why did the authors choose this data stream, with so few SF? Perhaps this is due to the frequency of these events compared to other sources. Can the authors compare the amount of SF events to that other publicly available datasets? What is the typical SNR level of seismic datasets on high-mountain areas and how does that compare to that of the application dataset? Did the authors consider adding some SF events from other datasets to increase the number of events in the training dataset? Perhaps the accuracy for SF on running windows during the test and application could be increased. These more specific details can be addressed in or after the introduction.

Another point related to the previous paragraph is that in the results, it is shown that only three out of eight events classified as SF by the algorithm are actual SF. This is the point highlighted in the abstract, which makes the reader lose confidence in the capabilities of the algorithm, and wonder if the chosen dataset was at all adequate for the goals of this study. My recommendation is that the authors bring most of the attention to the fact that, while false positives (FP) occur due to the low threshold chosen, only one false negative (FN) occurs, even in this sub-optimal conditions. Hence, even though accuracy is low, the ability to detect potential slope failures is high. Then, this can be linked to savings in visual monitoring time and confidence increase in the choices made by operators. In this way, deployment of this method can be highly valuable.

In general, the discussion is too shallow and does not provide a satisfactory explanation of several important points. The authors should also consider organizing it more clearly. I suggest discussing each point in a single paragraph, and referencing the sections and figures corresponding to each point being discussed. In sect. 5.1, the text could first address aspects related to the network and general dataset. Second, aspects related to training and testing the algorithm. Third, aspects related to misclassification of events in the 2019 application data. Finally, any other aspects. Most importantly, the underlying cause for the misclassifications is poorly discussed. It is mentioned that misclassification likely results from either similar frequency content or a low signal to noise ratio (e.g. in paragraphs 3 and 4 of sect. 5.1). However, the text should provide more insight. For example, could the noise be filtered out (i.e. is it located mostly in a different frequency band) and what would happen if that was done before classification? What are the specific characteristics of earthquake and SF signals that may make them too similar at low SNR, and can they be seen here? I would suggest extending Fig. 6, or making a new one, comparing 2019 SF with misclassified earthquakes and correctly identified earthquakes. How does the SNR, number of sensors, number of events, etc compare to what has been used in previous literature using random forest? What happens if 2 or 4 or another number of SF consecutive windows is used and what are the potential implications of the choices made here for future application settings? References should be added to support points such as these.

The conclusions are vague, do not focus on specific results, and are not strong enough. Three main findings of this analysis are that 1) near real-time automatic identification of SF is feasible (currently line 342); 2) sub-optimal network configuration, similar frequency content generated by different sources, and low SNR lead to false positives, requiring posterior manual data inspection; and 3) under sub-optimal conditions, this algorithm can outperform a 2-step STA/LTA detector and event classifier. I suggest presenting the main points that the authors consider most relevant first, written concisely, followed by the current second paragraph.

One important aspect of the classifier design is that any event related to a gravitational instability is considered part of the SF class. In lines 131-133, the text reads: "*We consider this assumption to be valid, as seismic source mechanisms of granular flows are similar and generate signals with similar characteristics*". This is a perfectly valid assumption, but the authors should describe the similarities. For example: is it the emergence of these type of signals? Do they display, at a given distance source-receiver, a particular energetic frequency band? Also, is there any kind of slope instability that generates signals especially similar to earthquakes? References should be added.

One final suggestion, that the authors can decide or not to follow, is to provide more explanation of the random forest model parameters. Currently, the authors write in line 227-228 that "*As a next step, the optimal model parameters (i.e., number of decision trees, number of features chosen for each tree, maximum tree depth, …) … are chosen …*", but no further explanation is provided. For readers interested in applying this methodology, it would be helpful (and likely little work for the authors) to add an appendix with a brief explanation of what does the randomized cross validation search consist of, and perhaps a table with the final model specifications (number of decision trees, features, maximum tree depth, degree of correlation between trees, and other relevant parameters).

**Technical corrections**

**Abstract**

line 8–9 The sentence starting with "*The presented method …*" is not clear. The authors should be clear in what they mean by "*facilitate data evaluation for stakeholders*". Perhaps something like *The presented method aims to reduce the amount of data requiring visual inspection, and facilitate detection of increased slope activity in a near real-time manner* could work better?

14–17 I would suggest rewriting the final lines of the abstract to emphasize what was mentioned above in the third paragraph of the specific comments. The modification in the abstract, from "*In total, …*" to the end, could be along the lines of *The algorithm correctly identifies three out of four actual slope failures. The missed slope failure was only [volume] and barely exceeded background noise, compared to the other three which ranged between [volume range]. Five additional events classified as SF are earthquakes with very similar spectral content and/or low SNR. Hence, we conclude that the training dataset and SNR limit the degree of accuracy that can be achieved, and that the method is suitable for supervised continuous near real-time seismic monitoring.*

**Introduction**

"*… affected by such instabilities*". Which type of instabilities? The authors should be specific, e.g. *affected by instabilities such as [add types]*.

25–26 "*Prediction of rockfall events is, due to lack of data and knowledge on relevant processes and triggering mechanisms, still not possible*". Does the text refer just to rockfalls or slope failures in general as in line 28? Also, references should be added to support this sentence at the end. Can the authors be more specific, in a couple of sentences, about what is not known?

"*However, an increase in activity …*" What type of activity? Should it be seismic activity?

"*Seismic signals generated by mass movements are typically emergent with dominant frequencies of 5-10 Hz and few or no distinguishable seismic phases*". Does this depend on the distance source-receiver? This should be clarified.

38–40 The authors should consider adding a figure to complement these lines and paragraph in general. The figure could contain a comparison of a typical seismic recording and spectrogram of the different types of mass movements named in the introduction. In relation to this, I suggest to be a bit more organized in the introduction: The text starts by referring to rock wall instabilities in general, followed by rockfalls, slope failures, mass movements and rock avalanches. It is not always clear to me if the authors are referring to a specific type of mass movement or if the comments apply to all types of mass movements. I understand that the goal is to present the state of the art as regards the importance of these events, monitoring techniques, and seismic signal characteristics. Should the text start with mass movements in general and then be narrowed down to the types most relevant in this analysis?

"*, that do not rely on an expert manually browsing through the data*". This is not needed, since the text already mentions "automated techniques".

48-49 I would suggest using past tense, when reporting about previous studies, to be consistent with the rest of the paragraph and text. For example, change "*…use a stochastic classifier…*" to *…used a stochastic classifier….* Alternatively, the authors can choose present tense and use it consistently.

48-49 Change "*…classify a variety of seismic sources, but focus on a regional…*" to *classify a variety of seismic sources. They focus(ed) on a regional…*

"*…on a regional scale with larger rockfall volumes…*" larger than what?

50-51 "*It has been shown, that HMMs successfully classify seismic signals on a continuous data stream*" Remove the comma after shown, and add references at the end of the sentence.

52-54 This sentence is not clear. Should it be changed to *Dammeier et al. (2016) compared the classification output with an earthquake catalog and suggested that, when using HMMs in actual operational settings, the on-duty operator…*?

57-58 Change "*…such as earthquakes and slope failures, a detection of only signals from one source mechanism with STA/LTA is impossible.*" into *…such as earthquakes and slope failures, limiting the detection to signals from a single source mechanism with STA/LTA is impossible.*

Change "*Aditionally, parameter selection is a tedious process…*" into "*Additionally, parameter selection for optimizing STA/LTA detection is a tedious process….*

71-72 Add references after "such processes.", i.e. at the end of the sentence.

73-77 For stronger writing, I suggest changing all these lines, i.e. from "*We show...*" until "an accurate model", into something like *Therefore, we use a sub-optimal seismic network of low-cost seismometers that does not allow for source location, nor particle motion analysis. We show that, even with a small number of recorded slope failures and low SNR, which increase the difficulty of training an accurate model, the detection of potential slope failures is still possible.* However, the authors should evaluate how to rewrite the last two paragraphs of the current introduction to address specific comments 1 and 2 (first 2 paragraphs).

78-82 Rewrite as necessary to address specific comments 1 and 2 (first 2 paragraphs).

**Study site and Instrumentation**

What landform type is Pizzo Cengalo? Is it a mountain? Please clarify.

88-89 Rewrite "*...as an acceleration of slope displacement was observed, as well as several smaller failure events...*" into *...because an acceleration of the slope displacement, as well as several smaller precursor failures, were observed...*

Refer to Fig. 1a when introducing the LERA network.

Important aspects related to the seismometers, such as the bandwidth/frequency range or the sampling frequency, should be listed.

93-94 Can the authors clarify this sentence? I.e. which stations and *how* amplitude differences at the stations are used to detect debris flows.

- To complete this section, the authors should consider adding a few sentences describing the geology of the site. What is the bedrock type at Pizzo Cengalo? Are there any studies that provide helpful measurements relating to the degree of fracturing of the rock matrix? Are there any scars, fresh surfaces or other indications of frequent activity at the site that the authors know of? How do these aspects compare to other instability-prone, widely studied areas?

**Methodology**

Consider changing "*...manually looking at the...*" to *...visually inspecting all of the recorded...*

Change "*...sample of the...*" to *...samples for the...*

Change "*...(Breiman, 2001) to classify a running window...*" to *...(Breiman, 2001), to classify different seismic sources using a running window...*

Could the authors clarify the meaning of the word *weak*, in relation to decision trees?

108-111 The sentence starting with "*We choose random forest...* is a bit too long and could use some more clarity. Perhaps it could be improved as follows: *We choose random forest, because (i.) it is a comprehensive machine learning algorithm that has been shown to outperform other algorithms, like support vector machines and boosting ensembles (...), and (ii.) it has already been successfully used to classify rock slope failures (...).*

Is this the same as the *variable importance* used to rank attributes (i.e. features) in Provost et al. (2017)? If so, why do the authors use different terminology? If not, perhaps the difference could be stated in a short sentence?

112-114 The description about what the impurity measures is not clear. Can the authors rewrite it to make it more accessible?

Add a comma after the word *algorithm*.

115-116 "*sci-kit learn*" is written differently depending on where it appears in the paper. What is the official name? I would suggest using the same spelling at all times.

Change "*from the LERA array*" to *from the 3-sensor LERA array.*

The authors mention that seismograms and spectrograms of these events (the four slope failures in 2019) are shown in Appendix A1. However, the caption of Fig. A1 refers to "*four additional slope failure events in 2018 used for training.*". So, either the caption does not correspond to the figure or the text should be clarified. Additionally, the authors should refer to Fig. A1 in the text.

122-124 I would suggest moving "*Here, we do not investigate source mechanisms and processes of seismogenic mass movements. The recorded signals are weak compared to other studies and thus not well suitable for such an endeavor.*" to the introduction, where the authors present their goals and scope.

Perhaps "*and close by stations*" should be changed to *and closest stations*?

Remove the comma after *shown* at the beginning of the line.

Fig. 2 should be mentioned in the text before Fig. 3.

157-164 Why do the authors choose these features and why 55? Is it to obtain a signal characterization as thorough as possible using individual metrics? Is it to maximize the number of features so that the degree of correlation between trees is minimized? Are there any drawbacks of using 55 features, some of which are similar, when compared to using a smaller number of more differentiated metrics? The fact that Provost et al. (2017) used a set of similar features is not enough to justify that this is the best choice for this study as well. Similar comments apply to the choice of the frequency bands.

I suggest considering to change the subsection title to *Imbalanced Data Set Handling*

I suggest changing "*highly disproportional*" to *significantly imbalanced*, and "*imposes*" to *poses*.

Add *especially* or *particularly* before *important* at the beginning of the line.

"*trainings*" should be changed to *training*, and references should be cited at the end of the sentence (after *algorithm*).

Add *i.e.* after "*overfitting,* "

Fig. 2c should perhaps be Fig. 2a if it is cited first.

In relation to the sentence "*The presented results for BRF use both class weights and undersampling.*" Can the authors refer to the corresponding results section and figures? Also, what weights were used and why? How was the training data undersampled?

Regarding the confusion matrix, the text says that the true label of each class is indicated in the rows, but in Fig. 2a it is in the columns.

Change "*...thresholds Fawcett (2006).*" to *...thresholds (Fawcett, 2006).*

I do not understand how Fig. 2a relates to the example given in the text.

Change "*...FPR of zero (0,1)*" to *...FPR of zero, i.e. coordinates (0,1),*

Change "*...this results simply in...*" to *...this simply results in...*

**Results**

Add figure reference after "*…oversampling techniques*", i.e. *oversampling techniques (Fig. 4).*

Specify that classical random forest means without any particular handling of imbalanced data: *…classical random forest, i.e. without modifications for handling imbalanced data.*

I think that the use of *catch*, in this and other lines, is not entirely correct. Should it be substituted by *detect*?

Should RF be SF? Otherwise, please clarify.

Could the authors indicate, here or elsewhere, how many time windows did each class contain in the test data?

Remove the last parenthesis in "The most discriminating features are presented in Fig. 5b). This typo comes up several times in the text.

246-247  In addition to Provost et al. (2017), table A1 should be referenced.

247-253  The authors talk about Fig. 5c, then 5b again, then 5c. I suggest to first describe the results presented in Fig. 5b, and then move on to Fig. 5c.

249-253  The authors should consider moving these two sentences to the discussion. Additionally, further comments should be provided. In the first sentence, the authors state that "*This is consistent with the fact that the windowing eliminates information from the entire waveform, amplitudes of signals strongly depend on emitted seismic energy and source receiver distance and the commonly observed differences in frequency patterns of noise signals, continuous seismic noise and other events (see Fig. 3)*" This sentence is too long and, in this form, lacks punctuation. The authors should consider subdividing or numbering each fact to make it clearer. Some questions that arise from the sentence, and should be discussed, are: Why is this the case? Are there other possibilities that could lead to this outcome? How does that compare to previous work?

The second sentence reads "*Figure 5c) shows however, that there is a large overlap between the classes, even for the most discriminating features, which highlights the necessity of a large number of features to distinguish the event type.*" Again, I suggest rewritting this into something like *However, the univariate distributions and correlations in Fig. 5c show large overlaps between the classes, even for the most discriminating features. This highlights the necessity of a large number of features to distinguish the event type.* Also, why do classes with large degree of overlap require many features for correct classification?

Why was an overlap of 26 seconds chosen?

259-261  Did the authors check the results with less or more consecutive windows? What were the differences as regards the number of misclassifications? Since this is an important choice that directly influences the accuracy of the classifier, this should be further discussed. How does the SNR compare to that of the 2018 data?

Change "*The misclassification likely results from a low signal to noise ratio (SNR), hence a probably relative small volume of the event, as all windows containing the event were classified as noise.*" to *The misclassification likely results from a low signal to noise ratio (SNR), i.e. a relative small volume of the event, as all windows containing the event were classified as noise (see sect. 5.1).* In the discussion, can the authors provide more insight regarding this false negative (FN)? This is an important point, since this event is the only FN. For example, is it possible to remove some of the noise in the signal, e.g. via filtering, and checking if this event would be correctly classified?

**Discussion**

- As a general comment, the text should refer to the results' subsections and figures that correspond to each point being discussed.

I suggest changing the title of subsection 5.1 to something like *Seismic network, data limitations and classifier performance.*

I suggest adding a few words at the end of the sentence "*Nevertheless, it is crucial to monitor the site.*" to explain why it is crucial to continue monitoring.

Change "*...the decrease in activity implies automatic detection...*" to *...the decrease in activity implies that automatic detection...*

279–291 For consistency, consider switching to past tense when reporting what has been done in the study.

Can the authors provide the actual values of TPR and FPR that they are referring to? I.e. what does *low* mean?

287–290 "*Generally, the problem of an imbalanced data set can be tackled by increasing the amount of training data in the minority class. A classifier trained for an area that is more active or has been monitored during a longer period is expected to give better results with higher accuracy. Additionally, the small number of events in the slope failure class can lead to overfitting, i.e., an insufficient generalization of the model.*" This has already been mentioned in the methods section. Instead, the text should be insightful regarding the reasons for misclassification in this specific dataset, and why this approach can work better/worse when using different datasets.

295–301 Although the spectral content of the earthquakes and SF at the site is very similar, the classifier is usually successful in differentiating earthquakes and SF if the SNR is not too low. Can the authors provide some explanation on the underlying reasons for this, if the classifier mostly relies on spectral features?

At the end of the paragraph, the authors clearly list the advantages of this approach as (1) eliminate false detections, (2) reduce parameter selection effort, and (3) create a more transparent system. The text should elaborate a bit more on the reasons for this, and, at least, specifically discuss these advantages in relation to (i.) the two-step methods using a STA/LTA detector in the time or frequency domain first (e.g. Provost et al. (2017)), and (ii.) HMMs (e.g. Dammeier et al. (2016)). Two other points that remain unclear at the end of the paragraph are, first, which are the methods that lead to false detections in other previous studies? and, second, why is this system more transparent?

Add comma before *however*.

306–309 As the authors note in the previous paragraph, quantifying the emergence of the signal seems to be a key parameter for confident automatic differentiation between earthquake and SF signals (e.g. Hibert et al., 2014; Provost et al., 2017). Unfortunately, this is not possible with the continuous window approach because the full waveform is required. For deployment purposes, perhaps this approach could benefit from a second step, admittedly introducing some time lag, in which each set of consecutive time windows classified as SF are collapsed into a pseudo-full waveform for a second evaluation with full waveform features. Do the authors consider this a viable strategy? What could be the potential advantages and inconveniences?

Refer to Fig. 5b and Table A1 after writing "*feature importance analysis*".

304-305 I suggest changing "*…values of time windows containing the slope failure signal…*" to *…values of time windows containing the misclassified slope failure signal…* for clarity.

This is an important section that strengthens this analysis. Perhaps the authors should consider changing the subsection title to something like *Classifier performance: constant time windows on continuous data vs STA/LTA detected events* for clarity.

Remove comma after *shown.* Also, since *several* is used, perhaps the authors can provide a few more references at the end of the sentence.

The authors should clarify that Provost et al. (2017) (and perhaps others) used the equivalent of a STA/LTA trigger in the frequency domain, based on spectrogram analysis (Helmstetter and Garambois, 2010). In this study, did the authors use a traditional STA/LTA or a modified version of it? Also, did the authors use additional features, i.e. those referring to the full waveform, to train and classify the detected events in the 2019 data?

As mentioned in the specific comments, the similarity between the SF and misclassified earthquake signals could be shown in an extension of Fig. 6 or a new figure. This would help illustrating and discussing the reasons for the misclassifications.

Consider using *In contrast,* or another contrast connector at the beginning of the sentence starting with "*The continuous approach…*"

333–334 The sentence "*Preliminary implementation of a fourth class called runoff with two days of increased water discharge (measured with gauges) found two more days of peak discharge*" is not clear. Specifically, found two more days of peak discharge than what?

334–335 Change "*Using the two step-method of STA/LTA, requires a second STA/LTA algorithm with its own parameters to detect these signals.*" to somethig like *Using the two-step method with an STA/LTA detector requires a second STA/LTA detector with its own parameters to detect these signals.*

"*…is potentially a low effort…*" should be changed to either *is potentially low effort* or, better, to *is potentially a low effort method* or similar.

**Conclusions**

339–345 The first two paragraphs should be rewritten into shorter, clearer points highlighting the main outcomes of this study (see my specific comments about the conclusions above).

Remove "*An added value is gained from a time consumption point of view:*"

**Figures**

Fig. A1 What are the magnitude units for the spectra in each plot? Can the authors define PSD? How was the spectrogram obtained (i.e. moving window length, overlap, etc)? This should all be provided, at least in the caption, and applies to the other figures showing spectra and spectrograms as well.

Note also that there is a discrepancy between the current caption in this figure, that refers to 2018 data, and what is understood from the text in line 120 (2019 data).

In the caption, change "*Bars show the total number of events with the transparent area…*" to *Bars show the total number of events with the lower opacity/higher transparency area…*

In the heatmap in Fig. 4a, I suggest using hotter colors to indicate larger values, simply because this tends to be the convention across the earth sciences.

**References**

F. Dammeier, J. R. Moore, C. Hammer, F. Haslinger, and S. Loew. Automatic detection of alpine rockslides in continuous seismic data using hidden Markov models. *Journal of Geophysical Research: Earth Surface*, 121(2):351–371, 2016. doi:10.1002/2015JF003647.

A. Helmstetter and S. Garambois. Seismic monitoring of séchilienne rockslide (french alps): Analysis of seismic signals and their correlation with rainfalls. *Journal of Geophysical Research: Earth Surface*, 115(F3), 2010. doi:10.1029/2009JF001532.

C. Hibert, A. Mangeney, G. Grandjean, C. Baillard, D. Rivet, N. M. Shapiro, C. Satriano, A. Maggi, P. Boissier, V. Ferrazzini, et al. Automated identification, location, and volume estimation of rockfalls at Piton de la Fournaise volcano. *Journal of Geophysical Research: Earth Surface*, 119(5):1082–1105, 2014. doi:10.1002/2013JF002970.

F. Provost, C. Hibert, and J.-P. Malet. Automatic classification of endogenous landslide seismicity using the random forest supervised classifier. *Geophysical Research Letters*, 44(1):113–120, 2017. doi:10.1002/2016GL070709.

---

## Author Comment (AC1) · 11 Sep 2020

Dear editor and reviewers,

Thank you for thorough reading and revisions of our manuscript "Near Real-Time Automated Classification of Seismic Signals of Slope Failures with Continuous Random Forests". Enclosed you will find a response to all reviewer comments on the manuscript. The most important changes to the manuscript are the following. i) We will add additional data from Illgraben, Switzerland and test the proposed method on both the Illgraben data set as well as the Bondo data set. ii) The final classifier will be trained on both data sets, which tends to improve the classifier. iii) More information on

the random forest parameters will be given. iv) The results and discussion section will be reorganized according to the reviewer comments. v) The discussion will be more thorough.

On the following pages, we provide a detailed point-by-point response to the reviewer's comments. Our replies are in blue. Most minor comments which are straightforward to implement (such as typos and rephrasing of sentences) are simply ticked off (using the ✓sign) without providing a response.

If you have any questions, we would be happy to answer them. We are looking forward to hearing from you about your decision.

Best regards,

Michaela Wenner

Comments of reviewer 1

**Comments to the Author
General questions:**

I read a study that has explored the potential of a machine learning algorithm to jointly detect and classify mass wasting and earthquake events from a small linear geophone array along a channel in the Swiss Alps. The study opens a new and timely avenue of "close to real time" hazard event warning by combining state of the art approaches in an arguably not optimally suited experimental setup. It discusses these drawbacks as well as different ways to account for them. The document is mostly well structured, provides adequate background, justification and motivation of the study. The applied/developed methodology is clearly described and can be digested without major ambiguities. The study is well placed in the scope of the journal and I am confident that after some modifications, it will be a valuable addition to the journal's portfolio.

Indeed, as the authors point out, the study is faced with suboptimal boundary

conditions. The most important drawbacks are i) network geometry (linear array with 20 m station spacing), ii) a lack of independent control on the hillslope events and iii), a striking event type imbalance (10ĔĘ1 hillslope events, 10ĔĘ2 earthquakes, 10ĔĘ3 noise cases). All these drawbacks are transparently mentioned, and their impact and counter measures are discussed in the text. Consequently, from a technical perspective, there is no reason to worry. However, it strikes my why this study design has been chosen to work with from the beginning. Why has this timely, rigorous and relevant study not been set up at a more suitable study site? There are many examples (cited in the text) where the network geometry is better (perhaps even in including a section of linear and densely spaced sensors to test the impact of such conditions, e.g. at the Sechilienne landslide), where there is excellent independent control on location, magnitude and to some degree the timing of hillslope activity, and where overall there are significantly more hillslope failure events that would lead to a less imbalanced data set? Somewhat, this excellent idea and study approach is vastly undersold due to the quality of the data.

The performance of RF or other ML techniques (e.g. HMMs) to distinguish between different seismogenic events has already been proven for ideal archived data (e.g. Provost et al., 2017). The original point of this study was to show that even though the network geometry and data availability are not ideal for the site, this method still gives valuable information on the occurrence of slope failure events. However, we understand that other data sets might have given better results in terms of the classification score. For this reason, we decided to include an event catalogue of seismic signals recorded with an array of eight stations at Illgraben, Switzerland. We will explore the performance of our proposed method on this data set, as well as have a look at how the classifier transfers from one site to the other. First tests have shown, that the performance of the classifier at Illgraben is similar to the performance shown for the Bondo site. Additionally, we found that a combination of both catalogues slightly improves the classification

results on the 2019 test data set of the Bondo test site.

Currently, a wider impact is impeded by the big question marks on the representativeness given that only a handful of slope failure events has been detected and this with a 270 % error (3 seismogram interpreted hillslope events versus 8 random forest-based hillslope events). Regarding the latter, while the abstract sounds quite confident (80 % prediction accuracy), the implementation of the approach does not. And it is a bit contradictory to claim the random forest approach would overcome manual inspection efforts to correctly classify an event, whereas in the discussion it becomes necessary to judge manually, which of the eight detected events is due to hillslope activity and which is an earthquake.

We realize that the representativeness is not ideal with such a small data set. We hope that by including the Illgraben data set, we can make better statements on this. Furthermore, we agree that the false positive rate is quite high, however compared to the more than a million data windows being classified in the 2019 data, we believe that 8 events falsely classified as slope failure events are reasonable. However, we should focus more on the false negatives instead of the false positives, as it is (arguably) more important to catch all events. Compared to other studies (e.g. Dammeier et al. 2016), the number of earthquakes detected slope failures is significantly smaller. We will rephrase statements in the text to highlight this better.

Long story short, I see two points that should receive more attention in the manuscript: i) a robust justification of the study site and experiment setup (Why working with an obviously unsuited network and missing event control?), and ii) a more thorough discussion of the classification errors, with due respect to the very small number of actual events and the resulting implications for the overall uncertainty.

1) See answer above 2) We will deepen the discussion on the above-mentioned points.

Regarding the classification quality part, one way that might be worth to explore is to use the hillslope events from the entire data set, not just the training subset. This of course only in the exploration of the classification quality (sections 4.1 and 4.2). The idea is to reduce the imbalance by increasing the number of hillslope events. In addition, this would shed some light on the actual impact of 5 versus 8 hillslope events.
We do not have a full labeled data set of 2019, but only the slope failure events, and checked the events that were falsely classified as slope failure events. This is why for an accurate testing of all classes, we are using parts of the 2018 data set. We agree however that the accuracy might increase by using the whole 2018 data set as training data and will test that.

**specific comments:**

The results section partly grades into a discussion. I recommend keeping these things separated, especially since there is a dedicated discussion section. Examples are l221-224, l249-254, l261-262, l268-269, l271-272.
We agree that the mentioned parts of the results section should be moved to the discussion section.

l20, I do not think it is necessary to use climate change as driver of this study. As in the abstract, it is sufficient to motivate by the mass movements, alone. But this is just a recommendation. No need to stick to that.
We would like to keep climate change as a driver for this study, as the threat for mountain communities will increase in the future, and simple and robust monitoring techniques will be the key for hazard monitoring and mitigation.

l34-35, check journal guidelines about order of references, here and throughout. Commonly, this is by date or author name, rather than apparently random order.
The references will be changed according to the journal guidelines.

l39-40, the larger amplitudes of slope failures must be compared to something. I assume you mean tremors. But the distance to the source will dominate the amplitude discussion. I suggest, to remove this misleading part of the sentence, it is of limited use, here. Overall, I am not sure the comparison of rock avalanches to tremors is a good one, especially in this journal and its readership.
We agree with this point and will remove the comparison to tremors from the sentence.

l48-54, well summarised. I suggest to pick that up in the discussion again, because like your routine the HMM approach also generates near-real time
classification of events. Thus, a verbal comparison of pros and cons of the two approaches is something the reader is interested in, and for good reason. Ideally, one would benchmark both approaches using the same input data, but I fear this is not feasible, here.

We agree that this is an important point, however, a benchmarking of both (or several more) approaches is not in the scope of this study. The pros and cons of both approaches will be added to the discussion.

l 55, the section about STA/LTA picking is a bit unfortunate, here. In the above paragraph you discuss detecting and classifying. Here you go back to just detecting. Would it not be more intuitive to first give a general introduction that defines and distinguishes detection and classification, and then elaborates on the different approaches to these two tasks? I suggest to write such a short introduction prior to l. 48. Then you can list the different approaches.

We agree that changing the order would be more intuitive and will do so in the revised manuscript.

✓ l76-77, that last sentence of the paragraph is actually results and discussion. I recommend to remove it here.

l78-80, in your scope, points a) and b) are not actually discussed and investigated. You do not write about decreased slope activity as a precursor of larger events or transitions of hillslope to channel activity. In fact, you cannot do this with only a hand full of events in total. I suggest to reword these points, here. Or simply collapse this paragraph with the above one after the corrections have been implemented.

We agree that this paragraph needs rewording, as the scope points in itself are not picked up again in the manuscript. We believe however, that our method

enables us to monitor an increase in slope activity or an early detection of hazardous events.

✓ l86, check SI unit conformity of volume numbers. Also see journal guidelines.

✓ l92, you may want to add more information about the loggers and recording frequencies, as well as on the installation of the sensors (surface, depth, coupling)?

l98, in the methods, I recommend adding the benchmark efforts that you discuss in section 5.2. This is a laudable and insightful test and it must be justified and described in the methods section.
We agree with the comment and will add a description of the STA/LTA benchmark to the methods section

✓ l101, check conformity of closing parenthesis in figure reference. Also, in other parts of the manuscript, this parenthesis is missing, check for correctness and consistency.

l134-135, this sentence kind of glances over a maybe important topic. Is there any way to show this more rigorously? I might suspect that i) local versus teleseismic earthquakes are quite distinct in terms of labeled features and ii) that smaller local quakes might be more similar to slope activity. Thus, could this lumping not be one reason for the result of 5 out of 8 hillslope event classifications being earthquakes? Usually, sentences that start with "After rigorous testing..." tend to hide potentially important subjective decisions instead of transparently showing the foundations of these decisions. Consequently, it would be good to be more transparent here, and show the effect of the lumped case versus for example two or three earthquake classes. Or at least to discuss

why for random forests it may be appropriate to stick to very small numbers of classes.

We do understand this criticism. It is true, that a large number of classes can improve prediction accuracy, due to a more accurate feature selection. However, as we are mostly interested in slope failures, we decided to keep the number of classes and the classification as "simple" as possible. Our first thought was also that keeping local, regional and teleseismic earthquakes separated would increase the prediction accuracy, this turned out to not be the case. We will follow the reviewer's suggestions and add plots showing the accuracy without lumping the classes in the appendix to back up this step.

l187, to account for the bias due to the imbalanced data set, can you not calculate the confusion matrix based on log-scaled numbers? I think in one of the Hammer HMM papers this has been done.

We think this is a good idea and will represent the confusion matrix in log-scaled numbers in the revised manuscript.

l216-217, why different colour schemes for the two matrices? It is not intuitive. No big deal but I may mention that it took me some thought to wonder why these different colours. Unless there is a reason (which should then be mentioned in the text/caption) I suggest to use the same colour scheme.

We used different colour schemes with one of them showing the AUC values and the other one is showing the 95% confidence interval. The different colour schemes were chosen to highlight the different meanings.

l220, reword, currently it reads as if RF and BRF are techniques at the same level as RF with US, OS and SMOTE. From the methods I read that US, OS and SMOTE are data manipulation steps prior to a subsequent RF classification,

no? Also, it would be good to actually discuss these findings later on (section 5). What does it mean that the imbalance countermeasures do not yield any improvement, but rather decrease the quality of the classification? What can we learn from that? What might be the reason?

Good point, we will reword that. Additionally, we understand that it might be misleading in the text, but the countermeasures do actually bring an improvement compared to an ordinary RF if we don't change the prediction threshold for RF. For this specific data set though, it seems that the accuracy for the slope failure class is highest if we do just that. From this we learn, that different techniques or improvements for an algorithm will not necessarily always give the best results for specific data sets. A few sentences on this will be added to the discussion part.

✓ l230, this number of 2 RF in the test data set comes out of the blue. Please revise and mention this at an appropriate place.

l258, the manual classification parameters must be defined in the methods (What are your classification judgements based on?). The image and radar methodology must be mentioned, as well. Also, since the catalogue is a key feature to validate your approach, I recommend to spend significantly more than just one short sentence on this topic, both in the methods description and the presentation of the resulting catalogue (a table or in the text).

Indeed, the catalogue is crucial for the testing. However, unfortunately not more information is available on that. We are in contact with the local stake holders who informed us on any reported events. We crosschecked radar and images, however not all events were caught, unfortunately, but no other events were caught either. We will elaborate on this in 1-2 additional sentences.

l262-263, I suggest you give more details here, in terms of description of the

events. It is only three failures, so there is space for that and it is important as the main goal of your study is to work out such events. Based on which criteria did you define these signals as hillslope failures? What are the event's properties? Also, in fig. 6, I only see one event and not all three. I suggest to plot the PSDs and seismograms also for the two other events, as in fig.6 c-e-g.

We will add the waveforms and spectrograms of the other events to the figure (or in a new figure) and describe the slope failures in the text.

l268-269, this is an unsupported statement. How are we to judge that this was an earthquake without seeing any data of it? Why do you think it is no hillslope event? Please present a PSD and seismogram as well as a more detailed description of the properties. This is the results section and it should present results sufficiently clear and exhaustive to allow you to draw conclusions from it.

We labeled this event as an earthquake as it shows a clear P and S-Wave arrival. However, we agree that a figure showing the event would be beneficial for the reader and will include one in the revised manuscript.

✓ l302, can you quantify this statement? What means high SNR, compared to what?

✓ l310, as mentioned above, this section should be motivated and described in the methods section, already. And its outcomes should be described in the results section, so that you can focus on the implications, here. Please revise.

✓ l311, delete comma after "shown".

l333, this is a valuable finding but strikingly out of context. Either include the runoff classification part from the beginning or leave it out (I recommend the

latter). Also, runoff appears to be a continuous feature rather than a comparably short lived event. In fact all PSDs of the manuscript show the seismic signature of water runoff. So why classifying it and how handling the case of two "events" occurring at the same time, such as runoff and rockfall?
We agree with the reviewer that discussing both runoff classification and "short" event detection in the same context seems contradictory. Accordingly, we will reword this discussion. Nevertheless, we prefer to leave the runoff part here, because though preliminary, our results suggest that the continuous RF classification can be applied beyond our study's scope.

✓ l342, revise this first sentence. Yes it is feasible, but with an error of 230 % (3 times right, 5 times wrong).

l345, rewrite "is a challenge that an imbalanced training data set enhances". Do you mean a challenge that is due to an imbalanced training data set? Or a challenge that may be solved by a less imbalanced training data set?
We understand this confusion and will reword accordingly.

✓ l349, manual inspection is not just advisable but crucial to account for the issue of misclassification, see comment two above. In the same line, replace "then" by "than" and "monitoring" by "inspecting".

l350-353, these are arm waving sentences. Either expand on this topic or leave it out. Currently this does not help the reader much. What is behind semi- and unsupervised ML algorithms, more specifically? Which specific drawbacks of the current approach would they solve? What are "unseen patterns"? I summary, I suggest not to mention this part, unless you find a way to explain its value in more detail.

We agree and will leave this part out.

Fig2c, value of that sketch is very limited. You may consider removing this panel. Fig.3, check font sizes, this is a really small font, hard to read. See journal guidelines on minimum size.
Fig 4, a and b homogenise colour schemes.
Fig 6, as mentioned above, also show other hillslope events, as well. Font on legend colour bar is too small.
All of these comments will be addressed (see comment above for color scale homogenization).

---

## Author Comment (AC2) · 11 Sep 2020

Dear editor and reviewers,

Thank you for thorough reading and revisions of our manuscript "Near Real-Time Automated Classification of Seismic Signals of Slope Failures with Continuous Random Forests". Enclosed you will find a response to all reviewer comments on the manuscript. The most important changes to the manuscript are the following. i) We will add additional data from Illgraben, Switzerland and test the proposed method on both the Illgraben data set as well as the Bondo data set. ii) The final classifier will be trained on both data sets, which tends to improve the classifier. iii) More information on

the random forest parameters will be given. iv) The results and discussion section will be reorganized according to the reviewer comments. v) The discussion will be more thorough.

On the following pages, we provide a detailed point-by-point response to the reviewer's comments. Our replies are in blue. Most minor comments which are straightforward to implement (such as typos and rephrasing of sentences) are simply ticked off (using the ✓ sign) without providing a response.

If you have any questions, we would be happy to answer them. We are looking forward to hearing from you about your decision.

Best regards,

Michaela Wenner

Comments of reviewer 2

**General comments:**

This paper presents the training, testing and application of a near-real time automated classifier of seismic signals. The text reports whether accurate identification of slope failure (SF) events is feasible using consecutive time windows on relatively noisy data from a small seismic network with little aperture. These are sub-optimal conditions for such an analysis, but so is often the case in areas with irregular topography and difficult access. Building on previous studies that relied on denser seismic networks and higher signal to noise ratios (SNR), the authors first train the algorithm with a dataset containing 1032 noise events, 3 SF and 17 earthquake signals. Then, the algorithm is tested on 441 noise events, 2 SF and 13 earthquake signals, using a running window with overlap on the continuous data stream; a prediction accuracy of 80% is achieved for SF. Finally, the algorithm is applied to 2019 data; three out of four

actual SF are correctly identified, but the algorithm also misidentifies five earthquakes as SF. The authors conclude that the presented method can be applied to near-real time continuous monitoring, and outline that this type of algorithms can substantially decrease operator time devoted to manual inspection of seismic data. This topic is well within the scope of NHESS and can be of interest to the communities of environmental seismology, natural hazards, early warning, and beyond.

Strong points of this study are (1) the introduction of a novel application of a random forest classifier using continuous data. The data stream is challenging, similar to what can be expected in hazardous, but increasingly transited and populated, high mountain settings; (2) performance of the presented algorithm is compared to that of a 2-step, well-established algorithm: A STA/LTA detector and classification using the full, i.e. STA/LTA-bounded, event. It is shown that, on the 2019 data, performance of the presented algorithm is superior to that of the 2-step algorithm; and (3) in this initial version, the text reads reasonably well and is, in general, adequately complemented by good-quality figures. Finally, the authors deserve credit for making publicly available the code used in this analysis.

In its current form, this paper suffers from (1) insufficient discussion of several important points, including the dataset, detection choices, misclassification of events, and advantages and potential issues of the chosen method. The authors are using a dataset with very few SF events to train, test and apply the algorithm, in addition to a low SNR of the application data. This results in low SF classification accuracy for the 2019 application data (3/8), so a careful and thorough discussion of these important points should be provided; and (2) several confusing points along the manuscript, due to unclear writing, insufficient explanation or both, which hinder comprehension of the choices and interpretations made. Therefore, before publication can be granted, I recommend a thoughtful review of the aspects detailed below.

**Specific comments:**

What is the goal of this study? Is the goal to detect slope failures (SFs) with high accuracy even when using running windows on a few number of relatively noisy channels? Is it to test whether any potential SF can be detected, to minimize operator time devoted to visually inspecting seismic data? Is it to test how much accuracy can be achieved with this algorithm applied to realistic noisy data and how it compares to the combination of a STA/LTA-type detector and a random forest classifier on the full event signal? The goal of the analysis should be clearly stated, not implicit in the text, since this is key for reading (and evaluating) the study. In line 78, the authors mention that the scope (do they mean goal?) of this study is to present an algorithm capable of the following "a) detect an increase in slope activity as an early warning for a plausible larger event in the near future and b) detect rock slope failures that possibly transition into hazardous debris flows early on, to enable down-stream communities to take action". However, the implications of a) and b) no longer appear in the text. Is an algorithm capable of a) always capable of b), and viceversa? Are there differences between signals of slope-related seismic activity before larger SF, and of SF that represent the beginning of debris flows? Is the trained algorithm capable of both a) and b)? If this is stated as the goal, then it should be discussed later. If this is just the general frame of the study, and the authors are merely naming two hazards that potentially generate precursor SF that can be detected, then the goal should be clearly listed and it should be ensured that there is no confusion.

We agree that the comments on the scope of the study might be misleading, as the points are not further discussed in the paper. We mean to say that despite the difficulties of a less than ideal network set up and noisy data with a small amount of training data, we are able to develop a tool that allows monitoring of slope failure events to catch trends of increasing slope activity or early warning of slope failure events. This will be clarified in the revised manuscript.

The instrumentation and dataset are very briefly presented in sect. 2 and 3. A careful explanation of the reasons for their choosing is required, because the network and dataset should be adequate for the study goals. Hence, the key issue here is to clearly justify that this network is adequate for this study. In the introduction, line 72-74, it is mentioned that "Previous local-scale approaches have used networks designed by experts and set-up as an array, ideal for monitoring such processes. However, due to cost and time constraints, this is not possible and not the case for most potential hazard sites". References should be added. Based on this, the authors state next that "We show that by adjusting our methodology to work with a network of low-cost seismometers with a sub-optimal network configuration, the detection of slope failures is still possible without post-processing". However, the following are still not clear: why did the authors choose this data stream, with so few SF? Perhaps this is due to the frequency of these events compared to other sources. Can the authors compare the amount of SF events to that other publicly available datasets? What is the typical SNR level of seismic datasets on high-mountain areas and how does that compare to that of the application dataset? Did the authors consider adding some SF events from other datasets to increase the number of events in the training dataset? Perhaps the accuracy for SF on running windows during the test and application could be increased. These more specific details can be addressed in or after the introduction.

We initially chose this setup to show that a satisfying classifier can be built on unideal conditions. However, we understand that from a scientific point of few, a richer data set will give more insight on the performance of the method. Hence, we decided to include an event catalogue of seismic signals recorded with an array of eight stations at Illgraben, Switzerland. This data set includes an tens of SF events, which provide a solid basis for a labeled training catalogue. We will explore the performance of our proposed method on this data set, as well as explore how the classifier transfers from on site to the other. First tests

have shown, that the performance of the classifier at Illgraben is similar to the performance shown for the Bondo site. Additionally, we found that a combination of both catalogues slightly improves the classification results on the 2019 test data set of the Bondo test site.

Another point related to the previous paragraph is that in the results, it is shown that only three out of eight events classified as SF by the algorithm are actual SF. This is the point highlighted in the abstract, which makes the reader lose confidence in the capabilities of the algorithm, and wonder if the chosen dataset was at all adequate for the goals of this study. My recommendation is that the authors bring most of the attention to the fact that, while false positives (FP) occur due to the low threshold chosen, only one false negative (FN) occurs, even in this sub-optimal conditions. Hence, even though accuracy is low, the ability to detect potential slope failures is high. Then, this can be linked to savings in visual monitoring time and confidence increase in the choices made by operators. In this way, deployment of this method can be highly valuable.
This is a good comment. We agree and will change the wording of these statements.

In general, the discussion is too shallow and does not provide a satisfactory explanation of several important points. The authors should also consider organizing it more clearly. I suggest discussing each point in a single paragraph, and referencing the sections and figures corresponding to each point being discussed. In sect. 5.1, the text could first address aspects related to the network and general dataset. Second, aspects related to training and testing the algorithm. Third, aspects related to misclassification of events in the 2019 application data. Finally, any other aspects. Most importantly, the underlying cause for the misclassifications is poorly discussed. It is mentioned that misclassification likely results from either similar frequency content or a low signal to

noise ratio (e.g. in paragraphs 3 and 4 of sect. 5.1). However, the text should provide more insight. For example, could the noise be filtered out (i.e. is it located mostly in a different frequency band) and what would happen if that was done before classification? What are the specific characteristics of earthquake and SF signals that may make them too similar at low SNR, and can they be seen here? I would suggest extending Fig. 6, or making a new one, comparing 2019 SF with misclassified earthquakes and correctly identified earthquakes. How does the SNR, number of sensors, number of events, etc compare to what has been used in previous literature using random forest? What happens if 2 or 4 or another number of SF consecutive windows is used and what are the potential implications of the choices made here for future application settings? References should be added to support points such as these.

We agree with the points and will adjust and add to our discussion accordingly.

The conclusions are vague, do not focus on specific results, and are not strong enough. Three main findings of this analysis are that 1) near real-time automatic identification of SF is feasible (currently line 342); 2) sub-optimal network configuration, similar frequency content generated by different sources, and low SNR lead to false positives, requiring posterior manual data inspection; and 3) under sub-optimal conditions, this algorithm can outperform a 2-step STA/LTA detector and event classifier. I suggest presenting the main points that the authors consider most relevant first, written concisely, followed by the current second paragraph.

Also here, we agree, and will restructure that conclusion.

One important aspect of the classifier design is that any event related to a gravitational instability is considered part of the SF class. In lines 131-133, the text reads: "We consider this assumption to be valid, as seismic source mechanisms of granular flows are similar and generate signals with similar characteristics". This is a perfectly valid assumption, but the authors should describe the similarities. For example: is it the emergence of these type of signals? Do they display, at a given distance source-receiver, a particular energetic frequency band? Also, is there any kind of slope instability that generates signals especially similar to earthquakes? References should be added.

The main similarities of the signals lie in the frequency band. Slope failure signals similar to earthquake signals are the ones of similar duration and impulse-like energy bursts that remind of phase arrivals in earthquake signals. We will discuss this in more detail.

One final suggestion, that the authors can decide or not to follow, is to provide more explanation of the random forest model parameters. Currently, the authors write in line 227-228 that "As a next step, the optimal model parameters (i.e., number of decision trees, number of features chosen for each tree, maximum tree depth, ...) ... are chosen ...", but no further explanation is provided. For readers interested in applying this methodology, it would be helpful (and likely little work for the authors) to add an appendix with a brief explanation of what does the randomized cross validation search consist of, and perhaps a table with the final model specifications (number of decision trees, features, maximum tree depth, degree of correlation between trees, and other relevant parameters).

We will add information on the RF parameters in the Appendix.

[Figure]

**Technical comments:**

✓ line 8–9 The sentence starting with "The presented method ..." is not clear. The authors should be clear in what they mean by "facilitate data evaluation for stakeholders". Perhaps something like The presented method aims to reduce the amount of data requiring visual inspection, and facilitate detection of increased slope activity in a near real-time manner could work better?

✓ 14–17 I would suggest rewriting the final lines of the abstract to emphasize what was mentioned above in the third paragraph of the specific comments. The modification in the abstract, from "In total, ..." to the end, could be along the lines of The algorithm correctly identifies three out of four actual slope failures. The missed slope failure was only [volume] and barely exceeded background noise, compared to the other three which ranged between [volume range]. Five additional events classified as SF are earthquakes with very similar spectral content and/or low SNR. Hence, we conclude that the training dataset and SNR limit the degree of accuracy that can be achieved, and that the method is suitable for supervised continuous near real-time seismic monitoring.

✓ 22 "... affected by such instabilities". Which type of instabilities? The authors should be specific, e.g. affected by instabilities such as [add types].

25–26 "Prediction of rockfall events is, due to lack of data and knowledge on relevant processes and triggering mechanisms, still not possible". Does the text refer just to rockfalls or slope failures in general as in line 28? Also, references should be added to support this sentence at the end. Can the authors be more specific, in a couple of sentences, about what is not known?
In this text we refer to rockfalls. Relevant references and explanations will be

added.

26 "However, an increase in activity ..." What type of activity? Should it be seismic activity?
Here we refer to slope failure activity. This will be rephrased in the text.

36 "Seismic signals generated by mass movements are typically emergent with dominant frequencies of 5-10 Hz and few or no distinguishable seismic phases". Does this depend on the distance source-receiver? This should be clarified.
It has been shown that the frequency content is affected by the source-receiver distance for debris flows, however the authors are not aware of a strong effect observed for rockfalls. Surely, higher frequencies experience higher attenuation effects with larger source-receiver distances. However, in the to us known literature, dominant frequencies always tend to be between 5 - 10 Hz. This will be clarified in the revised manuscript.

38–40 The authors should consider adding a figure to complement these lines and paragraph in general. The figure could contain a comparison of a typical seismic recording and spectrogram of the different types of mass movements named in the introduction. In relation to this, I suggest to be a bit more organized in the introduction: The text starts by referring to rock wall instabilities in general, followed by rockfalls, slope failures, mass movements and rock avalanches. It is not always clear to me if the authors are referring to a specific type of mass movement or if the comments apply to all types of mass movements. I understand that the goal is to present the state of the art as regards the importance of these events, monitoring techniques, and seismic signal characteristics. Should the text start with mass movements in general and then be narrowed down to the types most relevant in this analysis?
Following the reviewer's suggestion, we will reorganize this part of the text. On the other hand, we prefer not to give an overview figure with typical signal examples. Compiling catalogues of example waveforms has been discussed in the environmental seismological community for some time. A few efforts exist, but the challenge is always to capture the variety within a signal class. Providing too few examples poses the risk of oversimplification or of overemphasizing unimportant and misleading details.

✓ 47 ", that do not rely on an expert manually browsing through the data". This is not needed, since the text already mentions "automated techniques".

✓ 48-49 I would suggest using past tense, when reporting about previous studies, to be consistent with the rest of the paragraph and text. For example, change "...use a stochastic classifier..." to ...used a stochastic classifier.... Alternatively, the authors can choose present tense and use it consistently.

✓ 48-49 Change "...classify a variety of seismic sources, but focus on a regional..." to classify a variety of seismic sources. They focus(ed) on a regional...

49 "...on a regional scale with larger rockfall volumes..." larger than what?
Will be clarified in the revised manuscript.

✓ 50-51 "It has been shown, that HMMs successfully classify seismic signals on a continuous data stream" Remove the comma after shown, and add references at the end of the sentence.[-5pt]

✓ 52-54 This sentence is not clear. Should it be changed to Dammeier et al. (2016) compared the classification output with an earthquake catalog and suggested

that, when using HMMs in actual operational settings, the on-duty operator...?

✓ 57-58 Change "...such as earthquakes and slope failures, a detection of only signals from one source mechanism with STA/LTA is impossible." into ...such as earthquakes and slope failures, limiting the detection to signals from a single source mechanism with STA/LTA is impossible.

✓ 58 Change "Aditionally, parameter selection is a tedious process..." into "Additionally, parameter selection for optimizing STA/LTA detection is a tedious process....

✓ 71-72 Add references after "such processes.", i.e. at the end of the sentence.

✓ 73-77 For stronger writing, I suggest changing all these lines, i.e. from "We show..." until "an accurate model", into something like Therefore, we use a sub-optimal seismic network of low-cost seismometers that does not allow for source location, nor particle motion analysis. We show that, even with a small number of recorded slope failures and low SNR, which increase the difficulty of training an accurate model, the detection of potential slope failures is still possible. However, the authors should evaluate how to rewrite the last two paragraphs of the current introduction to address specific comments 1 and 2 (first 2 paragraphs).

✓ 78-82 Rewrite as necessary to address specific comments 1 and 2 (first 2 paragraphs).

✓ 84 What landform type is Pizzo Cengalo? Is it a mountain? Please clarify.

✓ 88-89 Rewrite "...as an acceleration of slope displacement was observed, as well as several smaller failure events..." into ...because an acceleration of the slope displacement, as well as several smaller precursor failures, were observed...

✓ 90 Refer to Fig. 1a when introducing the LERA network.

✓ 92 Important aspects related to the seismometers, such as the bandwidth/frequency range or the sampling frequency, should be listed.

93-94 Can the authors clarify this sentence? I.e. which stations and how amplitude differences at the stations are used to detect debris flows..
As debris flows approach a linear array of seismic stations the delayed increase in amplitude for down-stream stations is used as a detection parameter. This will be clarified in the text.

To complete this section, the authors should consider adding a few sentences describing the geology of the site. What is the bedrock type at Pizzo Cengalo? Are there any studies that provide helpful measurements relating to the degree of fracturing of the rock matrix? Are there any scars, fresh surfaces or other indications of frequent activity at the site that the authors know of? How do these aspects compare to other instability-prone, widely studied areas?
A few sentences will be added on the geology of the study site.

✓ 99 Consider changing "...manually looking at the..." to ...visually inspecting all of the recorded...

✓ 101 Change "...sample of the..." to ...samples for the...

✓ 102 Change "...(Breiman, 2001) to classify a running window..." to ...(Breiman, 2001), to classify different seismic sources using a running window...

103 Could the authors clarify the meaning of the word weak, in relation to decision trees?

A weak tree means that by only using one of those trees the classification result is not satisfying, i.e. has a low score. This is due to the fact, that it is only trained on a subset of the available data, as well as a subset of available features. We will specify this in the revised manuscript.

108-111 The sentence starting with "We choose random forest... is a bit too long and could use some more clarity. Perhaps it could be improved as follows: We choose random forest, because (i.) it is a comprehensive machine learning algorithm that has been shown to outperform other algorithms, like support vector machines and boosting ensembles (...), and (ii.) it has already been successfully used to classify rock slope failures (...).

We agree with the suggestions and will change the wording in the revised manuscript.

112 Is this the same as the variable importance used to rank attributes (i.e. features) in Provost et al. (2017)? If so, why do the authors use different terminology? If not, perhaps the difference could be stated in a short sentence?

We chose to use feature importance as this is, to our knowledge, the technical term widely used in machine learning literature. Variable importance does mean the same though, and is not wrong.

112-114 The description about what the impurity measures is not clear. Can the authors rewrite it to make it more accessible?

This will be clarified.

✓ 115 Add a comma after the word algorithm.

✓ 115-116 "sci-kit learn" is written differently depending on where it appears in the paper. What is the official name? I would suggest using the same spelling at all times.

✓ 118 Change "from the LERA array" to from the 3-sensor LERA array.

120 The authors mention that seismograms and spectrograms of these events (the four slope failures in 2019) are shown in Appendix A1. However, the caption of Fig. A1 refers to "four additional slope failure events in 2018 used for training.". So, either the caption does not correspond to the figure or the text should be clarified. Additionally, the authors should refer to Fig. A1 in the text.
This is a misunderstanding due to the sentence structure. Figure A1 indeed shows the waveforms and spectrograms of the 2018 events. Currently, not all 2019 events are shown in the manuscript. This will be added in the revised version.

122-124 I would suggest moving "Here, we do not investigate source mechanisms and processes of seismogenic mass movements. The recorded signals are weak compared to other studies and thus not well suitable for such an endeavor." to the introduction, where the authors present their goals and scope.
Thank you for this suggestion. We will change that.

✓ 126 Perhaps "and close by stations" should be changed to and closest stations?

✓ 135 Remove the comma after shown at the beginning of the line.

136 Fig. 2 should be mentioned in the text before Fig. 3.
We will change the order of the figures.

157-164 Why do the authors choose these features and why 55? Is it to obtain a signal characterization as thorough as possible using individual metrics? Is it to maximize the number of features so that the degree of correlation between trees is minimized? Are there any drawbacks of using 55 features, some of which are similar, when compared to using a smaller number of more differentiated metrics? The fact that Provost et al. (2017) used a set of similar features is not enough to justify that this is the best choice for this study as well. Similar comments apply to the choice of the frequency bands.
We chose the 55 features, as they have been proven significant in previous studies for an efficient and accurate classification. The frequency bands have been chosen based on the frequency content of interest. E.g. regional earthquake signals contain significant energy in lower frequency bands (e.g. 1 - 3 Hz) whereas slope failures generally contain low energy in the lower frequency bands. Starting this project, we have also tried a python package called TSFRESH (https://tsfresh.readthedocs.io/en/latest/), with which one can compute a large number of characteristics of time series. However, using these features did not improve the classification results. We will add a sentence on this.

✓ 165 I suggest considering to change the subsection title to Imbalanced Data Set Handling

✓ 167 I suggest changing "highly disproportional" to significantly imbalanced, and

"imposes" to poses.

✓ 171 Add especially or particularly before important at the beginning of the line.

✓ 172 "trainings" should be changed to training, and references should be cited at the end of the sentence (after algorithm).

✓ 176 Add i.e. after "overfitting, "

✓ 180 Fig. 2c should perhaps be Fig. 2a if it is cited first.

184 In relation to the sentence "The presented results for BRF use both class weights and undersampling." Can the authors refer to the corresponding results section and figures? Also, what weights were used and why? How was the training data undersampled?
The weights of the classes were used inversely proportional to the class frequency. The training data is undersampled in the majority classes to equalize the number of samples in the training data set. We just became aware of how if the training data set is anyway undersampled to an equal number of samples, class weights will be uniform. We will check this for the revised manuscript version.

190 Regarding the confusion matrix, the text says that the true label of each class is indicated in the rows, but in Fig. 2a it is in the columns.
We will change that.

✓ 193 Change "...thresholds Fawcett (2006)." to ...thresholds (Fawcett, 2006).

199 I do not understand how Fig. 2a relates to the example given in the text.
Numbers for the example will be included in the figure.

✓ 203 Change "...FPR of zero (0,1)" to ...FPR of zero, i.e. coordinates (0,1),

✓ 207 Change "...this results simply in..." to ...this simply results in...

✓ 215 Add figure reference after "...oversampling techniques", i.e. oversampling techniques (Fig. 4).

✓ 225 Specify that classical random forest means without any particular handling of imbalanced data: ...classical random forest, i.e. without modifications for handling imbalanced data.

✓ 225 I think that the use of catch, in this and other lines, is not entirely correct. Should it be substituted by detect?

230 Should RF be SF? Otherwise, please clarify.
Yes, indeed.

241 Could the authors indicate, here or elsewhere, how many time windows did each class contain in the test data?
We will include the total amount of testing windows in the revised manuscript.

✓ 242 Remove the last parenthesis in "The most discriminating features are presented in Fig. 5b). This typo comes up several times in the text.

✓ 246-247 In addition to Provost et al. (2017), table A1 should be referenced.

✓ 247-253 The authors talk about Fig. 5c, then 5b again, then 5c. I suggest to first describe the results presented in Fig. 5b, and then move on to Fig. 5c.

249-253 The authors should consider moving these two sentences to the discussion. Additionally, further comments should be provided. In the first sentence, the authors state that "This is consistent with the fact that the windowing eliminates information from the entire waveform, amplitudes of signals strongly depend on emitted seismic energy and source receiver distance and the commonly observed differences in frequency patterns of noise signals, continuous seismic noise and other events (see Fig. 3)" This sentence is too long and, in this form, lacks punctuation. The authors should consider subdividing or numbering each fact to make it clearer. Some questions that arise from the sentence, and should be discussed, are: Why is this the case? Are there other possibilities that could lead to this outcome? How does that compare to previous work? The second sentence reads "Figure 5c) shows however, that there is a large overlap between the classes, even for the most discriminating features, which highlights the necessity of a large number of features to distinguish the event type." Again, I suggest rewritting this into something like However, the univariate distributions and correlations in Fig. 5c show large overlaps between the classes, even for the most discriminating features. This highlights the necessity of a large number of features to distinguish the event type. Also, why do classes with large degree of overlap require many features for correct classification?
1. We agree and subdivide the sentence into several shorter ones and clarify the reviewer's concerns. The reasons we see for the outcome of the feature importance are a) losing information on the entire waveform and b) Amplitudes being strongly depended on the source-receiver distances as well as magnitudes of earthquakes or slope failure volumes. We cannot think of other possibilities

that could lead to this outcome. In previous work, especially the waveform features and network features have been shown to be the most important features. Both are, due to the continuous approach and the network set-up not available for our data set.

256 Why was an overlap of 26 seconds chosen?
The 26 seconds were chosen to keep the overlap consistent for the testing of all time windows (2/3 overlap). Other overlaps have not been tested.

259-261 Did the authors check the results with less or more consecutive windows? What were the differences as regards the number of misclassifications? Since this is an important choice that directly influences the accuracy of the classifier, this should be further discussed. How does the SNR compare to that of the 2018 data?
We are not sure to understand the question correctly. For the 2019 data we computed more the one million consecutive windows, which is more than for the 2018 training data. However, we do not have a manual catalog of the whole 2019 data, except for the slope failures that occurred, or the earthquakes that were mistakenly classified as slope failures. We will add information on the SNR of the 2019 and 2018 data.

✓ 274 I suggest changing the title of subsection 5.1 to something like Seismic network, data limitations and classifier performance.

276 I suggest adding a few words at the end of the sentence "Nevertheless, it is crucial to monitor the site." to explain why it is crucial to continue monitoring.
Good point, we will add a sentence here.

✓ 276 Change "...the decrease in activity implies automatic detection..." to ...the decrease in activity implies that automatic detection...

✓ 279–291 For consistency, consider switching to past tense when reporting what has been done in the study.

284 Can the authors provide the actual values of TPR and FPR that they are referring to? I.e. what does low mean?
We will add those values in the revised manuscript.

287–290 "Generally, the problem of an imbalanced data set can be tackled by increasing the amount of training data in the minority class. A classifier trained for an area that is more active or has been monitored during a longer period is expected to give better results with higher accuracy. Additionally, the small number of events in the slope failure class can lead to overfitting, i.e., an insufficient generalization of the model." This has already been mentioned in the methods section. Instead, the text should be insightful regarding the reasons for misclassification in this specific dataset, and why this approach can work better/worse when using different datasets.
We will add some sentences on that.

295–301 Although the spectral content of the earthquakes and SF at the site is very similar, the classifier is usually successful in differentiating earthquakes and SF if the SNR is not too low. Can the authors provide some explanation on the underlying reasons for this, if the classifier mostly relies on spectral features? At the end of the paragraph, the authors clearly list the advantages of this approach as (1) eliminate false detections, (2) reduce parameter selection effort, and (3) create a more transparent system. The text should elaborate a bit more on the

reasons for this, and, at least, specifically discuss these advantages in relation to (i.) the two-step methods using a STA/LTA detector in the time or frequency domain first (e.g. Provost et al. (2017)), and (ii.) HMMs (e.g. Dammeier et al. (2016)). Two other points that remain unclear at the end of the paragraph are, first, which are the methods that lead to false detections in other previous studies? and, second, why is this system more transparent?

1. We think that it is a combination of both the magnitude and the location of the earthquake that influences the misclassification as slope failure for our case. The one slope failure that was not classified as slope failure was classified as noise, due to its low SNR. 2. We will add this to the discussion.

✓ 296 Add comma before however.

306–309 As the authors note in the previous paragraph, quantifying the emergence of the signal seems to be a key parameter for confident automatic differentiation between earthquake and SF signals (e.g. Hibert et al., 2014; Provost et al., 2017). Unfortunately, this is not possible with the continuous window approach because the full waveform is required. For deployment purposes, perhaps this approach could benefit from a second step, admittedly introducing some time lag, in which each set of consecutive time windows classified as SF are collapsed into a pseudo-full waveform for a second evaluation with full waveform features. Do the authors consider this a viable strategy? What could be the potential advantages and inconveniences?

As the reviewer suggests, state transitioning of full waveform features could significantly enhance a detector performance. Data handling would be more involved and the classifying algorithm may be more complicated. However, we will pick up this point in the discussion as it seems a logical improvement and next step for our current method.

✓ 304 Refer to Fig. 5b and Table A1 after writing "feature importance analysis".

✓ 304-305 I suggest changing "...values of time windows containing the slope failure signal..." to ...values of time windows containing the misclassified slope failure signal... for clarity.

✓ 310 This is an important section that strengthens this analysis. Perhaps the authors should consider changing the subsection title to something like Classifier performance: constant time windows on continuous data vs STA/LTA detected events for clarity.

311 Remove comma after shown. Also, since several is used, perhaps the authors can provide a few more references at the end of the sentence.
More references will be added.

312 The authors should clarify that Provost et al. (2017) (and perhaps others) used the equivalent of a STA/LTA trigger in the frequency domain, based on spectrogram analysis (Helmstetter and Garambois, 2010). In this study, did the authors use a traditional STA/LTA or a modified version of it? Also, did the authors use additional features, i.e. those referring to the full waveform, to train and classify the detected events in the 2019 data?
We will clarify that. In this study, we also used an STA/LTA trigger in the frequency domain and included features on the whole waveform.

325 As mentioned in the specific comments, the similarity between the SF and misclassified earthquake signals could be shown in an extension of Fig. 6 or a new figure. This would help illustrating and discussing the reasons for the misclassifications.

We will show the as slope failure signals and misclassified signals in a new Figure

✓ 330 Consider using In contrast, or another contrast connector at the beginning of the sentence starting with "The continuous approach..."

333–334 The sentence "Preliminary implementation of a fourth class called runoff with two days of increased water discharge (measured with gauges) found two more days of peak discharge" is not clear. Specifically, found two more days of peak discharge than what?
We refer to two additional days in the 2019 data, as we also trained with two days of the 2019 data. We will rephrase that in the revised manuscript.

✓ 334–335 Change "Using the two step-method of STA/LTA, requires a second STA/LTA algorithm with its own parameters to detect these signals." to somethig like Using the two-step method with an STA/LTA detector requires a second STA/LTA detector with its own parameters to detect these signals.

✓ 336 "...is potentially a low effort..." should be changed to either is potentially low effort or, better, to is potentially a low effort method or similar.

339–345 The first two paragraphs should be rewritten into shorter, clearer points highlighting the main outcomes of this study (see my specific comments about the conclusions above).
We will rewrite this part.

✓ 346 Remove "An added value is gained from a time consumption point of view:"

Fig. A1 What are the magnitude units for the spectra in each plot? Can the authors define PSD? How was the spectrogram obtained (i.e. moving window length, overlap, etc)? This should all be provided, at least in the caption, and applies to the other figures showing spectra and spectrograms as well. Note also that there is a discrepancy between the current caption in this figure, that refers to 2018 data, and what is understood from the text in line 120 (2019 data).
We will add the specifics in the caption. The text in line 120 was written in a misleading way and actually refers to the 2018 data.

✓ Fig 3 In the caption, change ”Bars show the total number of events with the transparent area...” to Bars show the total number of events with the lower opacity/higher transparency area...

Fig 4 In the heatmap in Fig. 4a, I suggest using hotter colors to indicate larger values, simply because this tends to be the convention across the earth sciences.
Good point, we will change that.

---

## Author Response (AR2)

Eidgenössische Technische Hochschule Zürich
Swiss Federal Institute of Technology Zurich

**Laboratory of
Hydraulics, Hydrology and Glaciology**

ETH Zurich
Michaela Wenner
PhD Student
HIA D 54.1
Hönggerbergring 26
8093 Zurich, Switzerland

Phone  +41 44 632 33 16
wenner@vaw.baug.ethz.ch

To: The editor of Natural Hazards and Earth System Sciences

Zurich, December 7, 2020

**Revision of manuscript NHESS-2020-200**

Dear editor and reviewers,

Thank you for your quick revisions of the revised version of our manuscript "Near Real-Time Automated Classification of Seismic Signals of Slope Failures with Continuous Random Forests". Enclosed you will find a response to the reviewer comments on the manuscript. The most important changes to the manuscript are: We clarified the motivation of the study in the abstract and the introduction and we restructured the section on the one-step vs. two-step method.

On the following pages, we provide a detailed point-by-point response to the reviewer's comments. Our replies are in blue. Most minor comments which are straightforward to implement (such as typos and rephrasing of sentences) are simply ticked off (using the ✓sign) without providing a response.

If you have any questions, we would be happy to answer them. We are looking forward to hearing from you about your decision.

Best regards,

M. Wenner

Michaela Wenner

Comments of reviewer 1

The revised version of the text improved in many occasions as the authors did a thorough job. I consider the addition of a further study site a valuable asset that gives the study a bolder shape. Below I give a few comments and suggestions that involve in my view minor changes to the text. Thus, I would not insist on seeing the manuscript another time unless the editor sees an urgent need to decide otherwise.

In general, the addition of the Illgraben site should prompt a (slight) change in the motivation and pitch of the manuscript. My impression is that you could motivate the study by saying that the approach has been shown to work under ideal conditions, which are in many use cases rarely met, and that you want to explore the effects of i) less than ideal network geometries but high SNR events (Pizzo Cengalo), ii) well suited network geometry but low SNR events (Illgraben), and iii) imbalanced training data sets (both? sites). This would give a proper justification of your study design. The new pitch should be better reflected in the abstract (apart from just mentioning you work on two different data sets) and also the end of the introduction.
We agree and tried to adjust to that in the abstract as well as in the introduction.

✓ L 7, change "earthquakes" to "earthquake"

✓ L 8, mention the names of the two locations (Illgraben, Pizzo Cengalo) when listing the data sets

L 9–12, level of detail is imbalanced between the two study site. Consider more equal representation of the results between the two sites.
Very good point. We equaled the information on both data sets by only including information on the accuracy of the slope failure class.

✓ L 24, LaTEX bibliography issue

✓ L 37, define what you mean with "large events". Is that for example millions of $m^3$? Or "events that affect entire slopes"? Depending on who reads the article, there may be different personal definitions about "large events".

✓ L 41, I would add the Ekstrom and Stark reference at the end of the sentence, because this is a nice example of continuous monitoring of large (global) areas.

L 76–80, this is where the adjustments of scope/pitch/motivation could be implemented (see my general comment above).
We included and rewrote a few sentences at this point.

✓ L 85, does this last sentence not make more sense if it were to occur further up, just after the sentence ending in the middle on line 83?

✓ L 185, change "train" to "training"

L192, "does not allow" … "nor does it allow", "nor" requires use of "neither" before. Consider rewriting.
We considered this but decided to leave the sentence the original way. Including neither somehow interrupts the flow of the sentence, and the preceding "not" actually allows the usage of nor without neither, as far as we are aware.

L 357–359, can this guessed implication be tested? Otherwise, consider to explicitly mention that you cannot test this. Currently, the "might" is a bit odd and leaves us unsatisfied about what the sentence should imply.
This can be tested over time by adding future events to the data set, as we expect a performance enhancement then. We added a sentence here.

✓ L 392–396, move this part to the results and focus here on the actual discussion of these results, on explaining why the two-step approach performed less well.

L 400–408, this part appears out of context with what is discussed before and after, and especially with what the section title implies.
We see that point and tried to embed the paragraph in a better way. The point of this part was to draw attention to how the continuous approach sacrifices some rather important features, but the continuous approach still seems to perform better. We added a few sentences on that.

L 430, "break-off", do you mean "detachment" (as process) or "detachment zone/site/area" (as spatial description)?
Good point. We changed the wording to "detachment zone"..

L 434, "bad weather conditions during snowfall periods", please clarify what "bad" means. Can this be an artifact?
We understand the confusion and changed the wording. In this context we meant "bad weather conditions" as an obscured view of the camera, and therefore no useful pictures we could use to identify events.

✓ L 437, add "snow" before "avalanches"

L 476–480, I am not quite sure if I can follow this argumentation. According to l 331, a classifier trained on both sites increased the minority class TP (?) value by 10 %. Thus, you might want to relax the statement of "greatly improve" (l 479) a bit, especially when considering the small amount of underlying slope movement events.
We agree with this statement and changed the wording to a more moderate conclusion.

[revised manuscript text omitted]